# Momentum Tracking: Momentum Acceleration for Decentralized Deep Learning on Heterogeneous Data

## Abstract

SGD with momentum acceleration is one of the key components for improving the performance of neural networks. For decentralized learning, a straightforward approach using momentum acceleration is Distributed SGD (DSGD) with momentum acceleration (DSGDm). However, DSGDm performs worse than DSGD when the data distributions are statistically heterogeneous. Recently, several studies have addressed this issue and proposed methods with momentum acceleration that are more robust to data heterogeneity than DSGDm, although their convergence rates remain dependent on data heterogeneity and decrease when the data distributions are heterogeneous. In this study, we propose Momentum Tracking, which is a method with momentum acceleration whose convergence rate is proven to be independent of data heterogeneity. More specifically, we analyze the convergence rate of Momentum Tracking in the standard deep learning setting, where the objective function is non-convex and the stochastic gradient is used. Then, we identify that it is independent of data heterogeneity for any momentum coefficient $\beta \in [0, 1)$. Through image classification tasks, we demonstrate that Momentum Tracking is more robust to data heterogeneity than the existing decentralized learning methods with momentum acceleration and can consistently outperform these existing methods when the data distributions are heterogeneous.

## 1 Introduction

Neural networks have achieved remarkable success in various fields such as image processing (Simonyan & Zisserman, 2015; Chen et al., 2020) and natural language processing (Devlin et al., 2019). To train neural networks, we need to collect large amounts of training data. However, because of privacy concerns, it is often difficult to collect large amounts of data such as medical images on one server. In such scenarios, decentralized learning has attracted significant attention because it allows us to train neural networks without aggregating all the data onto one server. Recently, decentralized learning has been studied from various perspectives, including data heterogeneity (Tang et al., 2018b; Esfandiari et al., 2021), communication compression (Tang et al., 2018a; Lu & De Sa, 2020; Liu et al., 2021; Takezawa et al., 2022a), and network topologies (Ying et al., 2021).

One of the key components for improving the performance of neural networks is SGD with momentum acceleration (SGDm). Whereas SGD updates the model parameters using a stochastic gradient, SGDm updates the model parameters using the moving average of the stochastic gradient, which is called the momentum. Because SGDm can accelerate convergence and improve generalization performance, SGDm has become an indispensable tool, enabling neural networks to achieve high accuracy (He et al., 2016). Recently, SGDm has been improved in many studies, and methods such as Adam (Kingma & Ba, 2015) and RAdam (Liu et al., 2020a) have been proposed.

In decentralized learning, the straightforward approach to using the momentum is Distributed SGD (DSGD) with momentum acceleration (DSGDm) (Gao & Huang, 2020). When the data distributions held by each node (i.e., the server) are statistically homogeneous, DSGDm works well and can improve the performance as well as SGDm (Lin et al., 2021). However, in real-world decentralized learning settings, the data distributions may be heterogeneous (Hsieh et al., 2020). In such cases, DSGDm performs worse than DSGD (i.e., without momentum acceleration) (Yuan et al., 2021).

Table 1: Comparison of the convergence rates. In the "Data-Heterogeneity" column, "✓" indicates that the convergence rate is independent of data heterogeneity, and "(✓)" indicates that it is independent, but there is no discussion about data heterogeneity either theoretically or experimentally. In the "Momentum," "Stochastic," and "Non-Convex" columns, "✓" respectively indicates that the method is accelerated using momentum, the convergence rate is provided when the stochastic gradient is used, and the convergence rate is provided when the objective function is non-convex.

| | Data-Heterogeneity | Momentum | Stochastic | Non-Convex |
|---|:---:|:---:|:---:|:---:|
| DSGD (Lian et al., 2017) | | | ✓ | ✓ |
| Gradient Tracking (Koloskova et al., 2021) | ✓ | | ✓ | ✓ |
| DSGDm (Gao & Huang, 2020) | | ✓ | ✓ | ✓ |
| QG-DSGDm (Lin et al., 2021) | | ✓ | ✓ | ✓ |
| DecentLaM (Yuan et al., 2021) | | ✓ | ✓ | ✓ |
| ABm (Xin & Khan, 2020) | (✓) | ✓ | | |
| GTAdam (Carnevale et al., 2022) | (✓) | ✓ | | |
| **Momentum Tracking (our work)** | ✓ | ✓ | ✓ | ✓ |

This is because, when the data distributions are heterogeneous and we use the momentum instead of the stochastic gradient, each model parameter is updated in further different directions and drifts away more easily. As a result, the convergence rate of DSGDm falls below that of DSGD. To address this issue, Lin et al. (2021) and Yuan et al. (2021) modified the update rules of the momentum in DSGDm and proposed methods that are more robust to data heterogeneity than DSGDm. However, their convergence rates remain dependent on data heterogeneity, and our experiments revealed that their performance are degraded when the data distributions are strongly heterogeneous (Sec. 4).

Data heterogeneity for decentralized learning has been well studied from both experimental and theoretical perspectives (Hsieh et al., 2020; Koloskova et al., 2020). Subsequently, many methods including Gradient Tracking (Lorenzo & Scutari, 2016; Nedić et al., 2017) have been proposed and it has been shown that their convergence rates do not depend on data heterogeneity (Tang et al., 2018b; Vogels et al., 2021; Koloskova et al., 2021). However, these studies considered only the case where the momentum was not used, and it remains unclear whether these methods are robust to data heterogeneity when the momentum is applied.

In the convex optimization literature, Xin & Khan (2020) and Carnevale et al. (2022) proposed combining Gradient Tracking with momentum or Adam and analyzed the convergence rates. However, they considered only the case where the objective function is strongly convex and the full gradient is used, which does not hold in the standard deep learning setting, where the objective function is non-convex and only the stochastic gradient is accessible. Hence, their convergence rates are still unknown in standard deep learning settings, and it remains unclear whether their convergence rates are independent of data heterogeneity. Furthermore, they did not discuss data heterogeneity, either theoretically or experimentally.

In this work, we propose a decentralized learning method with momentum acceleration, which we call **Momentum Tracking**, whose convergence rate is proven to be independent of data heterogeneity in the standard deep learning setting. More specifically, we provide the convergence rate of Momentum Tracking in a setting in which the objective function is non-convex and the stochastic gradient is used. Then, we identify that the convergence rate of Momentum Tracking is independent of data heterogeneity for any momentum coefficient $\beta \in [0, 1)$. In Table 1, we compare the convergence rate of Momentum Tracking with those of existing methods. To the best of our knowledge, Momentum Tracking is the first decentralized learning method with momentum acceleration whose convergence rate has been proven to be independent of data heterogeneity in the standard deep learning setting. Experimentally, we demonstrate that Momentum Tracking is more robust to data heterogeneity than the existing decentralized learning methods with momentum acceleration and can consistently outperform these existing methods when the data distributions are heterogeneous.

## 2 PRELIMINARIES AND RELATED WORK

### 2.1 DECENTRALIZED LEARNING

Let $G = (V, E)$ be an undirected graph that represents the underlying network topology, where $V$ denotes the set of nodes and $E$ denotes the set of edges. Let $N := |V|$ be the number of nodes, and we label each node in $V$ by a set of integers $\{1, 2, \cdots, N\}$ for simplicity. We define $\mathcal{N}_i := \{j \in$

$V \mid (i,j) \in E\}$ as the set of neighbor nodes of node $i$ and define $\mathcal{N}_i^+ := \mathcal{N}_i \cup \{i\}$. In decentralized learning, node $i$ has a local data distribution $\mathcal{D}_i$ and local objective function $f_i : \mathbb{R}^d \to \mathbb{R}$, and can communicate with node $j$ if and only if $(i,j) \in E$. Then, decentralized learning aims to minimize the average of the local objective functions as follows:

$$\min_{\boldsymbol{x} \in \mathbb{R}^d} \left[ f(\boldsymbol{x}) := \frac{1}{N} \sum_{i=1}^N f_i(\boldsymbol{x}) \right], \quad f_i(\boldsymbol{x}) := \mathbb{E}_{\xi_i \sim \mathcal{D}_i} \left[ F_i(\boldsymbol{x}; \xi_i) \right],$$

where $\boldsymbol{x}$ is the model parameter, $\xi_i$ is the data sample that follows $\mathcal{D}_i$, and local objective function $f_i(\boldsymbol{x})$ is defined as the expectation of $F_i(\boldsymbol{x}; \xi_i)$ over data sample $\xi_i$. In the following, $\nabla F_i(\boldsymbol{x}; \xi_i)$ and $\nabla f_i(\boldsymbol{x}) := \mathbb{E}_{\xi_i \sim \mathcal{D}_i}[\nabla F_i(\boldsymbol{x}; \xi_i)]$ denote the stochastic and full gradient respectively.

Distributed SGD (DSGD) (Lian et al., 2017) is one of the most well-known algorithms for decentralized learning. Formally, the update rules of DSGD are defined as follows:

$$\boldsymbol{x}_i^{(r+1)} = \sum_{j \in \mathcal{N}_i^+} W_{ij} \left( \boldsymbol{x}_j^{(r)} - \eta \nabla F_j(\boldsymbol{x}_j^{(r)}; \xi_j^{(r)}) \right), \tag{1}$$

where $\eta > 0$ is the step size and $W_{ij} \in [0,1]$ is the weight of edge $(i,j)$. Let $\boldsymbol{W} \in [0,1]^{N \times N}$ be the matrix whose $(i,j)$-element is $W_{ij}$ if $(i,j) \in E$ and 0 otherwise. In general, a mixing matrix is used for $\boldsymbol{W}$ (i.e., $\boldsymbol{W} = \boldsymbol{W}^\top$, $\boldsymbol{W}\boldsymbol{1} = \boldsymbol{1}$, and $\boldsymbol{W}^\top \boldsymbol{1} = \boldsymbol{1}$). Lian et al. (2018) extended DSGD in the case where each node communicates asynchronously and analyzed the convergence rate. Koloskova et al. (2020) analyzed the convergence rate of DSGD when the network topology changes over time. These results revealed that the convergence rate of DSGD decreases and the performance is degraded when the data distributions held by each node are statistically heterogeneous. This is because the local gradients $\nabla f_i$ are different across nodes and each model parameter $\boldsymbol{x}_i$ tends to drift away when the data distributions are heterogeneous. To address this issue, $D^2$ (Tang et al., 2018b), Gradient Tracking (Lorenzo & Scutari, 2016; Nedić et al., 2017), and primal-dual algorithms (Niwa et al., 2020; 2021; Takezawa et al., 2022b) were proposed to correct the local gradient $\nabla f_i$ to the global gradient $\nabla f$. As a different approach, Vogels et al. (2021) proposed a novel averaging method to prevent each model parameter $\boldsymbol{x}_i$ from drifting away. It has been shown that the convergence rates of these methods do not depend on data heterogeneity and do not decrease, even when the data distributions held by each node are statistically heterogeneous. However, these methods do not consider the case in which momentum is used.

## 2.2 Momentum Acceleration

The methods with momentum acceleration were originally proposed by Polyak (1964), and SGD with momentum acceleration (SGDm) has achieved successful results in training neural networks (Simonyan & Zisserman, 2015; He et al., 2016; Wang et al., 2020b). In decentralized learning, a straightforward approach to using the momentum is DSGD with momentum acceleration (DSGDm) (Gao & Huang, 2020). The update rules of DSGDm are defined as follows:

$$\boldsymbol{u}_i^{(r+1)} = \beta \boldsymbol{u}_i^{(r)} + \nabla F_i(\boldsymbol{x}_i^{(r)}; \xi_i^{(r)}), \tag{2}$$

$$\boldsymbol{x}_i^{(r+1)} = \sum_{j \in \mathcal{N}_i^+} W_{ij} \left( \boldsymbol{x}_j^{(r)} - \eta \boldsymbol{u}_j^{(r+1)} \right), \tag{3}$$

where $\boldsymbol{u}_i$ is the local momentum of node $i$ and $\beta \in [0,1)$ is a momentum coefficient. In addition, several variants of DSGDm were studied by Yu et al. (2019); Assran et al. (2019); Wang et al. (2020a); Singh et al. (2021). When the data distributions held by each node are statistically homogeneous, DSGDm works well and can improve the performance as well as SGDm. However, when the data distributions are statistically heterogeneous, DSGDm leads to poorer performance than DSGD. This is because when the data distributions held by each node are statistically heterogeneous (i.e., $\nabla f_i$ varies significantly across nodes), the difference in the updated value of the model parameter across the nodes (i.e., $\eta \boldsymbol{u}_i$) is amplified by the momentum (Lin et al., 2021).

To address this issue, Yuan et al. (2021) and Lin et al. (2021) proposed methods to modify the update rules of the momentum in DSGDm, called DecentLaM and QG-DSGDm, respectively. They further experimentally demonstrated that these methods are more robust to data heterogeneity than DSGDm. However, their convergence rates have been shown to still depend on data heterogeneity and decrease when the data distributions are heterogeneous.

### 2.3 GRADIENT TRACKING

One of the most well-known methods whose convergence rate does not depend on data heterogeneity is Gradient Tracking (Lorenzo & Scutari, 2016). Whereas DSGD exchanges only the model parameter $\boldsymbol{x}_i$, Gradient Tracking exchanges the model parameter $\boldsymbol{x}_i$ and local (stochastic) gradient $\nabla f_i$ and then updates the model parameters while estimating global gradient $\nabla f$. Nedić et al. (2017) and Qu & Li (2018) analyzed the convergence rate of Gradient Tracking when the objective function is (strongly) convex and the full gradient is used. Pu & Nedic (2021) analyzed the convergence rate when the objective function is strongly convex and the stochastic gradient is used. Recently, Koloskova et al. (2021) analyzed the convergence rates of Gradient Tracking in a standard deep learning setting, where the objective function is non-convex and the stochastic gradient is used. There is also a line of research to combine Gradient Tracking with variance reduction methods (Xin et al., 2022). They showed that the convergence rate of Gradient Tracking does not depend on data heterogeneity. However, these studies only consider the case without momentum acceleration, and the convergence analysis for Gradient Tracking with momentum acceleration has not been explored thus far in the aforementioned studies.

In the convex optimization literature, Xin & Khan (2020) and Carnevale et al. (2022) proposed a combination of Gradient Tracking and the momentum or Adam (Kingma & Ba, 2015). However, they only considered the case where the objective function is strongly convex and the full gradient is used. The convergence rate is still unclear in the standard deep learning setting, where the objective function is non-convex and the stochastic gradient is used. Furthermore, there is no discussion about data heterogeneity in these studies, either theoretically or experimentally.

## 3 PROPOSED METHOD

In this section, we propose **Momentum Tracking**, which is a decentralized learning method with momentum acceleration whose convergence rate is proven to be independent of the data heterogeneity in the standard deep learning setting.

### 3.1 SETUP

We assume that the following four standard assumptions hold:

**Assumption 1.** *There exists a constant $f^\star > -\infty$ that satisfies $f(\boldsymbol{x}) \geq f^\star$ for all $\boldsymbol{x} \in \mathbb{R}^d$.*

**Assumption 2.** *There exists a constant $p \in (0, 1]$ that satisfies for all $\boldsymbol{x}_1, \cdots, \boldsymbol{x}_N \in \mathbb{R}^d$,*

$$\|\boldsymbol{X}\boldsymbol{W} - \bar{\boldsymbol{X}}\|_F^2 \leq (1 - p)\|\boldsymbol{X} - \bar{\boldsymbol{X}}\|_F^2, \tag{4}$$

*where $\boldsymbol{X} := (\boldsymbol{x}_1, \cdots, \boldsymbol{x}_N) \in \mathbb{R}^{d \times N}$ and $\bar{\boldsymbol{X}} := \frac{1}{N}\boldsymbol{X}\mathbf{1}\mathbf{1}^\top$.*

**Assumption 3.** *There exists a constant $L > 0$ that satisfies for all $i \in V$ and $\boldsymbol{x}, \boldsymbol{y} \in \mathbb{R}^d$,*

$$\|\nabla f_i(\boldsymbol{x}) - \nabla f_i(\boldsymbol{y})\| \leq L\|\boldsymbol{x} - \boldsymbol{y}\|. \tag{5}$$

**Assumption 4.** *There exists a constant $\sigma^2$ that satisfies for all $i \in V$ and $\boldsymbol{x}_i \in \mathbb{R}^d$,*

$$\mathbb{E}_{\xi_i \sim \mathcal{D}_i}\|\nabla F_i(\boldsymbol{x}_i; \xi_i) - \nabla f_i(\boldsymbol{x}_i)\|^2 \leq \sigma^2. \tag{6}$$

Assumptions 1, 2, 3, and 4 are commonly used for decentralized learning algorithms (Lian et al., 2017; Yu et al., 2019; Koloskova et al., 2021; Lin et al., 2021). Additionally, the following assumption, which represents data heterogeneity, is commonly used in the convergence analysis of decentralized learning algorithms (Lian et al., 2017; Yu et al., 2019; Lin et al., 2021).

**Assumption 5.** *There exists a constant $\zeta^2$ that satisfies for all $\boldsymbol{x} \in \mathbb{R}^d$,*

$$\frac{1}{N}\sum_{i=1}^{N}\|\nabla f_i(\boldsymbol{x}) - \nabla f(\boldsymbol{x})\|^2 \leq \zeta^2.$$

Under Assumption 5, the convergence rates of DSGD (Lian et al., 2017), DSGDm (Gao & Huang, 2020; Yuan et al., 2021), QG-DSGDm (Lin et al., 2021), and DecentLaM (Yuan et al., 2021) were

shown to be dependent on data heterogeneity $\zeta^2$ and decrease as $\zeta^2$ increases. By contrast, in Sec. 3.3, we prove that Momentum Tracking converges without Assumption 5 and the convergence rate is independent of data heterogeneity $\zeta^2$. In addition, we do not assume the convexity of the objective functions $f(\boldsymbol{x})$ and $f_i(\boldsymbol{x})$. Therefore, $f(\boldsymbol{x})$ and $f_i(\boldsymbol{x})$ are potentially non-convex functions (e.g., the loss functions of neural networks).

## 3.2 MOMENTUM TRACKING

In this section, we propose **Momentum Tracking**, which is robust to data heterogeneity and accelerated by the momentum. The update rules of Momentum Tracking are defined as follows:

$$\boldsymbol{u}_i^{(r+1)} = \beta \boldsymbol{u}_i^{(r)} + \nabla F_i(\boldsymbol{x}_i^{(r)}; \xi_i^{(r)}), \tag{7}$$

$$\boldsymbol{x}_i^{(r+1)} = \sum_{j \in \mathcal{N}_i^+} W_{ij} \boldsymbol{x}_j^{(r)} - \eta \left( \boldsymbol{u}_i^{(r+1)} - \boldsymbol{c}_i^{(r)} \right), \tag{8}$$

$$\boldsymbol{c}_i^{(r+1)} = \sum_{j \in \mathcal{N}_i^+} W_{ij} \left( \boldsymbol{c}_j^{(r)} - \boldsymbol{u}_j^{(r+1)} \right) + \boldsymbol{u}_i^{(r+1)}, \tag{9}$$

where $\beta \in [0, 1)$ is a momentum coefficient. The pseudo-code for Momentum Tracking is presented in Sec. A. In Momentum Tracking, $\boldsymbol{c}_i$ corrects the local momentum $\boldsymbol{u}_i$ to the global momentum $\frac{1}{N} \sum_j \boldsymbol{u}_j$ and prevents each model parameter $\boldsymbol{x}_i$ from drifting, even when the data distributions are statistically heterogeneous (i.e., the local momentum $\boldsymbol{u}_i$ varies significantly across nodes).

Because Momentum Tracking is equivalent to Gradient Tracking when $\beta = 0$, Momentum Tracking is a simple extension of Gradient Tracking. Hence, when $\beta = 0$, it has been shown that the convergence rate of Momentum Tracking is independent of data heterogeneity $\zeta^2$ (Koloskova et al., 2021). However, because data heterogeneity is amplified when the momentum is used instead of the stochastic gradient (i.e., $\beta > 0$) (Lin et al., 2021; Yuan et al., 2021), it is unclear whether the convergence rate of Momentum Tracking is independent of data heterogeneity $\zeta^2$ for any $\beta \in [0, 1)$ or for only a restricted range of $\beta$. In Sec. 3.3, we provide the convergence rate of Momentum Tracking and prove that it is independent of data heterogeneity $\zeta^2$ for any $\beta \in [0, 1)$.

## 3.3 CONVERGENCE ANALYSIS

Under Assumptions 1, 2, 3, and 4, Theorem 1 provides the convergence rate of Momentum Tracking in the standard deep learning setting. All proofs are presented in Sec. D.

**Theorem 1** (Convergence Rate in Non-Convex Setting). *Suppose that Assumptions 1, 2, 3, and 4 hold, each model parameter $\boldsymbol{x}_i$ is initialized with the same parameters, and both $\boldsymbol{u}_i$ and $\boldsymbol{c}_i$ are initialized as $\frac{1}{1-\beta}(\nabla F_i(\boldsymbol{x}_i^{(0)}; \xi_i^{(0)}) - \frac{1}{N}\sum_{j=1}^{N} \nabla F_j(\boldsymbol{x}_j^{(0)}; \xi_j^{(0)}))$. Then, for any $\beta \in [0, 1)$ and $R \geq 1$, there exists a step size $\eta$ such that the average parameter $\bar{\boldsymbol{x}} := \frac{1}{N}\sum_{i=1}^{N} \boldsymbol{x}_i$ generated by Eqs. (7-9) satisfies*

$$\frac{1}{R} \sum_{r=0}^{R-1} \mathbb{E} \left\| \nabla f(\bar{\boldsymbol{x}}^{(r)}) \right\|^2 \tag{10}$$

$$\leq \mathcal{O} \left( \sqrt{\frac{r_0 \sigma^2 L}{NR}} + \left( \frac{r_0^2 \sigma^2 L^2}{p^4 R^2 (1-\beta)} \left( 1 + \frac{p\beta^2}{1-\beta} \right) \right)^{\frac{1}{3}} + \frac{Lr_0}{(1-\beta)p^2 R} \sqrt{1 + \frac{\beta^2}{(1-\beta^2)^3 p}} \right),$$

*where $r_0 := f(\bar{\boldsymbol{x}}^{(0)}) - f^\star$.*

**Remark 1.** *Combinations of Gradient Tracking with the momentum or Adam have also been proposed by Xin & Khan (2020) and Carnevale et al. (2022). However, they considered only the setting in which the objective function is strongly convex and the full gradient is used. By contrast, our study focuses on the deep learning setting. Hence, our proof strategies are completely different from those in these previous studies, and Theorem 1 provides the convergence rate in the setting where the objective function is non-convex and the stochastic gradient is used.*

**Remark 2.** *The convergence rate of Gradient Tracking in the standard deep learning setting was provided by Koloskova et al. (2020). However, they did not consider the case where the momentum is used, and it is not trivial to provide the convergence rate of Momentum Tracking from the results in this previous work.*

### 3.4 DISCUSSION

**Comparison with Gradient Tracking:** Theorem 1 indicates that the convergence rate of Momentum Tracking does not depend on data heterogeneity $\zeta^2$ for any $\beta \in [0, 1)$ and does not decrease even when the data distributions are statistically heterogeneous (i.e., $\zeta^2 > 0$). Therefore, Theorem 1 indicates that Momentum Tracking is theoretically robust to data heterogeneity for any $\beta \in [0, 1)$. Although Momentum Tracking is a simple extension of Gradient Tracking, our work is the first to identify that the combination of Gradient Tracking and the momentum converges without being affected by data heterogeneity $\zeta^2$ for any $\beta \in [0, 1)$ in the standard deep learning setting.

Because the convergence rate of Momentum Tracking Eq. (10) is optimal when $\beta = 0$, Theorem 1 does not show that the convergence rate is improved by using the momentum. However, the convergence rates of DSGDm and QG-DSGDm provided by Gao & Huang (2020) and Lin et al. (2021) are also optimal when $\beta = 0$. Moreover, they do not provide theoretical results that are consistent with the experimental results that the convergence rates are improved when $\beta > 0$. As in these studies, we experimentally demonstrate that convergence is accelerated when $\beta > 0$ in Sec. 4 and leave for future work to show the theoretical benefits of using $\beta > 0$.

**Comparison with Existing Algorithms with Momentum Acceleration:** Next, we compare the convergence rate of Momentum Tracking with those of existing decentralized learning algorithms with momentum acceleration: DSGDm (Gao & Huang, 2020), DecentLaM (Yuan et al., 2021), and QG-DSGDm (Lin et al., 2021). Here, we only show the convergence rate of QG-DSGDm, but the same discussion holds for the other methods. The convergence rate of QG-DSGDm is as follows:

**Theorem 2** ((Lin et al., 2021)). *Suppose that Assumptions 1, 2, 3, and 4 hold, and Assumption 5 also holds. Then, for any $\beta \in [0, \frac{p}{21+p}]$ and $R \geq 1$, there exists a step size $\eta$ such that the average parameter $\bar{x} := \frac{1}{N} \sum_i x_i$ generated by QG-DSGDm satisfies*[1]

$$\frac{1}{R} \sum_{r=0}^{R-1} \mathbb{E} \left\| \nabla f(\bar{x}^{(r)}) \right\|^2 \leq \mathcal{O} \left( \sqrt{\frac{r_0 \sigma^2 L}{NR}} + \left( \frac{r_0^2 L^2 (\zeta^2 + \sigma^2)}{p^2 R^2} \right)^{\frac{1}{3}} + \frac{L r_0}{R} \left( \frac{1}{p} + \frac{1}{1-\beta} + \frac{\beta}{(1-\beta)^3} \right) \right),$$

*where $r_0 := f(\bar{x}^{(0)}) - f^\star$.*

Data heterogeneity $\zeta^2$ appears in the second term, and the convergence rate of QG-DSGDm depends on data heterogeneity $\zeta^2$. Therefore, the convergence rate of QG-DSGDm decreases when the data distributions held by each node are statistically heterogeneous. By contrast, the convergence rate of Momentum Tracking Eq. (10) does not depend on data heterogeneity $\zeta^2$. Therefore, Momentum Tracking is more robust to data heterogeneity than QG-DSGDm. Because the convergence rates of DSGDm and DecentLaM also depend on data heterogeneity $\zeta^2$, the same discussion holds for DSGDm and DecentLaM. Hence, Momentum Tracking is more robust to data heterogeneity than these methods. To the best of our knowledge, Momentum Tracking is the first decentralized learning method with momentum acceleration whose convergence rate has been proven to be independent of data heterogeneity $\zeta^2$ in the standard deep learning setting.

Next, we discuss the range of $\beta$. The convergence rates of QG-DSGDm and DecentLaM provided by Lin et al. (2021) and Yuan et al. (2021) hold only when the range of $\beta$ is restricted. For instance, Theorem 2 assumes that $\beta \leq \frac{p}{21+p} (< 0.05)$. However, these restrictions on the range of $\beta$ do not hold in practice. (Typically, $\beta$ is set to 0.9.) Therefore, the convergence rates of QG-DSGDm and DecentLaM are unclear in such practical cases. By contrast, Theorem 1 can provide the convergence rate of Momentum Tracking that holds for any $\beta \in [0, 1)$.

**Comparison with SGDm:** Next, we compare the convergence rate of Momentum Tracking with that of SGDm. In a setting where the objective function is non-convex and the stochastic gradient is used, SGDm has been proven to converge to the stationary point with $\mathcal{O}(1/\sqrt{R})$ (Yan et al., 2018; Liu et al., 2020b). By contrast, Theorem 1 indicates that if the number of rounds $R$ is sufficiently large, Momentum Tracking converges with $\mathcal{O}(1/\sqrt{NR})$. Therefore, Momentum Tracking can achieve a linear speedup with respect to the number of nodes $N$, which is a common and important property in decentralized learning methods (Lian et al., 2018; Koloskova et al., 2020).

---

[1] For simplicity, we set the additional hyperparameter $\mu$ for QG-DSGDm to $\beta$.

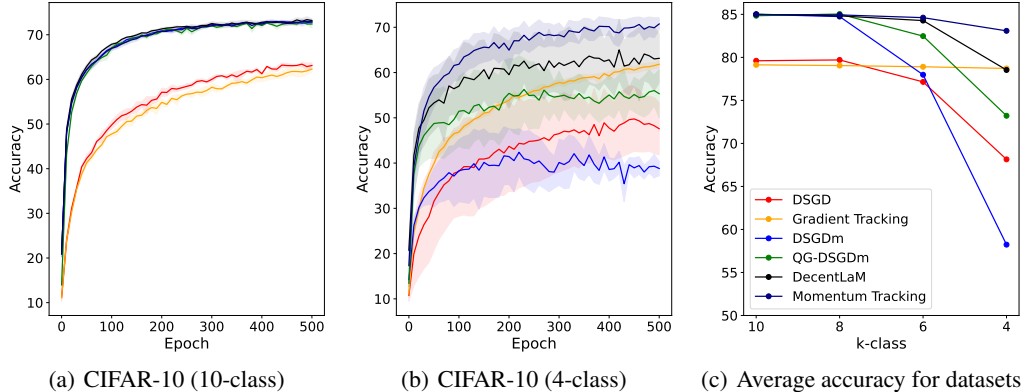

(a) CIFAR-10 (10-class)     (b) CIFAR-10 (4-class)     (c) Average accuracy for datasets

Figure 1: (a) Learning curve on CIFAR-10 with LeNet in the 10-class (i.e., homogeneous) setting. We evaluated the test accuracy per 10 epochs. (b) Learning curve in the 4-class (i.e., heterogeneous) setting. (c) Average test accuracy for all datasets (i.e., FashionMNIST, SVHN, and CIFAR-10).

## 4 EXPERIMENT

In this section, we present the results of an experimental evaluation of Momentum Tracking and demonstrate that Momentum Tracking is more robust to data heterogeneity than the existing decentralized learning methods with momentum acceleration. In this section, we focus on test accuracy, and more detailed evaluation about the convergence rate is presented in Sec. C.4.

### 4.1 SETUP

**Comparison Methods:** (1) DSGD (Lian et al., 2017): the method described in Sec. 2.1; (2) DSGDm (Gao & Huang, 2020): the method described in Sec. 2.2; (3) QG-DSGDm (Lin et al., 2021): a method in which the update rule of the momentum in DSGDm is modified to be more robust to data heterogeneity than DSGDm; (4) DecentLaM (Yuan et al., 2021): a method in which the update rule of the momentum in DSGDm is modified to be more robust to data heterogeneity; (5) Gradient Tracking (Nedić et al., 2017): a method without momentum acceleration that is robust to data heterogeneity; (6) Momentum Tracking: the proposed method described in Sec. 3.

**Dataset and Model:** We evaluated Momentum Tracking using three 10-class image classification tasks: FashionMNIST (Xiao et al., 2017), SVHN (Netzer et al., 2011), and CIFAR-10 (Krizhevsky, 2009). Following the previous work (Niwa et al., 2020), we distributed the data to nodes such that each node was given data of randomly selected $k$ classes. When $k = 10$, the data distributions held by each node can be regarded as statistically homogeneous. When $k < 10$, the data distributions are regarded as statistically heterogeneous. We evaluated the comparison methods by setting $k$ to $\{4, 6, 8, 10\}$ and changing data heterogeneity. Note that a smaller $k$ indicates that the data distributions are more heterogeneous. For the neural network architecture, we used LeNet (LeCun et al., 1998) with group normalization (Wu & He, 2018) in Sec. 4.2. In Sec. 4.3, we present more detailed evaluation by varying the neural network architecture (e.g., VGG-11 (Simonyan & Zisserman, 2015) and ResNet-34 (He et al., 2016)). For each comparison method, we used 10% of the training data for validation and individually tuned the step size. For DSGDm, QG-DSGDm, DecentLaM, and Momentum Tracking, we set $\beta$ to 0.9. All experiments were repeated using three different seed values, and we report their averages. More detailed hyperparameter settings are presented in Sec E.

**Network Topology and Implementation:** In Secs. 4.2 and 4.3, we present the results of setting the underlying network topology to a ring consisting of eight nodes (i.e., $N = 8$). In Sec. C.1, we present more detailed evaluation by varying the network topology. All comparison methods were implemented using PyTorch and run on eight GPUs (NVIDIA RTX 3090).

### 4.2 EXPERIMENTAL RESULTS

Table 2 lists the test accuracy for FashionMNIST, SVHN, and CIFAR-10. Fig. 1 (a) and (b) present the learning curves for CIFAR-10 and Fig. 1 (c) presents the average test accuracy for all datasets.

Table 2: Test accuracy on FashionMNIST, SVHN, and CIFAR-10 with LeNet. "$k$-class" means that each node has only the data of randomly selected $k$ classes. Bold font means the highest accuracy.

| | FashionMNIST | | | |
| --- | --- | --- | --- | --- |
| | 10-class | 8-class | 6-class | 4-class |
| DSGD | $85.6 \pm 0.49$ | $85.6 \pm 0.41$ | $82.7 \pm 1.12$ | $78.1 \pm 1.56$ |
| Gradient Tracking | $85.0 \pm 0.49$ | $85.4 \pm 0.26$ | $85.0 \pm 0.37$ | $84.9 \pm 0.22$ |
| DSGDm | $89.5 \pm 0.15$ | $89.3 \pm 0.21$ | $82.1 \pm 3.23$ | $68.7 \pm 5.02$ |
| QG-DSGDm | $\mathbf{89.6 \pm 0.10}$ | $\mathbf{89.5 \pm 0.47}$ | $86.9 \pm 1.59$ | $80.8 \pm 2.94$ |
| DecentLaM | $89.5 \pm 0.14$ | $89.3 \pm 0.36$ | $\mathbf{89.2 \pm 0.41}$ | $84.3 \pm 3.05$ |
| Momentum Tracking | $89.5 \pm 0.36$ | $89.4 \pm 0.05$ | $88.9 \pm 0.47$ | $\mathbf{86.8 \pm 1.56}$ |

| | SVHN | | | |
| --- | --- | --- | --- | --- |
| | 10-class | 8-class | 6-class | 4-class |
| DSGD | $90.1 \pm 0.17$ | $89.5 \pm 0.61$ | $87.6 \pm 1.94$ | $78.8 \pm 8.55$ |
| Gradient Tracking | $90.1 \pm 0.30$ | $89.8 \pm 0.38$ | $89.8 \pm 0.39$ | $89.4 \pm 0.47$ |
| DSGDm | $\mathbf{92.6 \pm 0.35}$ | $92.4 \pm 0.19$ | $88.1 \pm 4.38$ | $67.2 \pm 9.69$ |
| QG-DSGDm | $92.5 \pm 0.22$ | $\mathbf{92.5 \pm 0.17}$ | $90.9 \pm 1.67$ | $83.5 \pm 7.14$ |
| DecentLaM | $92.4 \pm 0.21$ | $92.2 \pm 0.39$ | $92.0 \pm 0.48$ | $88.2 \pm 4.75$ |
| Momentum Tracking | $\mathbf{92.6 \pm 0.32}$ | $92.4 \pm 0.40$ | $\mathbf{92.3 \pm 0.23}$ | $\mathbf{91.7 \pm 0.53}$ |

| | CIFAR-10 | | | |
| --- | --- | --- | --- | --- |
| | 10-class | 8-class | 6-class | 4-class |
| DSGD | $63.1 \pm 0.60$ | $64.1 \pm 0.52$ | $61.2 \pm 1.16$ | $47.6 \pm 5.77$ |
| Gradient Tracking | $62.3 \pm 0.73$ | $62.0 \pm 0.80$ | $61.9 \pm 0.58$ | $61.8 \pm 0.82$ |
| DSGDm | $72.9 \pm 0.41$ | $72.5 \pm 0.20$ | $63.8 \pm 6.24$ | $38.8 \pm 1.61$ |
| QG-DSGDm | $72.4 \pm 0.87$ | $\mathbf{73.1 \pm 0.16}$ | $69.6 \pm 2.42$ | $55.3 \pm 5.30$ |
| DecentLaM | $\mathbf{73.2 \pm 0.36}$ | $72.9 \pm 0.14$ | $71.7 \pm 1.10$ | $63.1 \pm 5.43$ |
| Momentum Tracking | $72.9 \pm 0.59$ | $73.0 \pm 0.49$ | $\mathbf{72.6 \pm 0.41}$ | $\mathbf{70.7 \pm 1.38}$ |

**Comparison of Momentum Tracking and Gradient Tracking:** First, we discuss the results of Momentum Tracking and Gradient Tracking. Table 2 and Fig. 1 indicate that Momentum Tracking achieves a higher accuracy faster than Gradient Tracking and outperforms Gradient Tracking in all settings. When the data distributions are homogeneous (i.e., 10-class), Momentum Tracking outperforms Gradient Tracking by $5.8\%$ on average. When the data distributions are heterogeneous (e.g., 4-class), Momentum Tracking outperforms Gradient Tracking by $4.4\%$ on average. Therefore, the results show that Momentum Tracking can consistently outperform Gradient Tracking regardless of data heterogeneity.

**Comparison of Momentum Tracking and DSGDm:** Next, we discuss the results of Momentum Tracking and DSGDm. The results show that when the data distributions are homogeneous (i.e., 10-class), Momentum Tracking and DSGDm are comparable and outperform DSGD and Gradient Tracking. However, when the data distributions are heterogeneous (e.g., 4-class), the test accuracy of DSGDm decreases even more than that of DSGD, and DSGDm underperforms DSGD by $9.9\%$ on average. By contrast, the results indicate that Momentum Tracking consistently outperforms DSGD and Gradient Tracking by $14.9\%$ and $4.4\%$ respectively when the data distributions are heterogeneous. The results indicate that Momentum Tracking is more robust to data heterogeneity than DSGDm and outperforms DSGDm by $24.9\%$ on average.

**Comparison of Momentum Tracking, QG-DSGDm, and DecentLaM:** When the data distributions are homogeneous (i.e., 10-class), Momentum Tracking, QG-DSGDm, and DecentLaM are comparable and outperform DSGD and Gradient Tracking. By contrast, when the data distributions are heterogeneous (e.g., 4-class), Momentum Tracking consistently outperforms QG-DSGDm and DecentLaM by $9.9\%$ and $4.5\%$ respectively, whereas QG-DSGDm and DecentLaM are more robust to data heterogeneity than DSGDm. Hence, these results are consistent with our theoretical analysis, as discussed in Secs. 3.3 and 3.4.

Table 3: Test accuracy on CIFAR-10 with VGG-11 and ResNet-34. "$k$-class" indicates that each node has only the data of randomly selected $k$ classes, and bold font indicates the highest accuracy.

| | CIFAR-10 + VGG-11 | | | CIFAR-10 + ResNet-34 | | |
|---|---|---|---|---|---|---|
| | 10-class | 4-class | 2-class | 10-class | 4-class | 2-class |
| DSGD | $89.7 \pm 0.15$ | $85.8 \pm 2.39$ | $70.3 \pm 2.73$ | $93.4 \pm 0.05$ | $87.5 \pm 3.27$ | $64.1 \pm 0.54$ |
| Gradient Tracking | $88.1 \pm 0.19$ | $86.0 \pm 0.45$ | $82.9 \pm 0.13$ | $85.1 \pm 1.07$ | $81.2 \pm 0.73$ | $75.9 \pm 0.41$ |
| DSGDm | $\mathbf{92.2 \pm 0.09}$ | $77.3 \pm 4.05$ | $39.6 \pm 5.92$ | $95.8 \pm 0.26$ | $79.0 \pm 3.69$ | $27.7 \pm 2.83$ |
| QG-DSGDm | $92.0 \pm 0.04$ | $89.4 \pm 1.04$ | $77.8 \pm 1.96$ | $95.6 \pm 0.37$ | $94.0 \pm 1.02$ | $77.4 \pm 3.13$ |
| DecentLaM | $92.1 \pm 0.09$ | $\mathbf{90.9 \pm 0.65}$ | $85.2 \pm 0.67$ | $\mathbf{95.9 \pm 0.04}$ | $\mathbf{95.2 \pm 0.51}$ | $89.2 \pm 2.26$ |
| Momentum Tracking | $91.9 \pm 0.06$ | $\mathbf{90.9 \pm 0.60}$ | $\mathbf{87.0 \pm 0.48}$ | $95.0 \pm 0.13$ | $94.4 \pm 0.52$ | $\mathbf{89.9 \pm 0.73}$ |

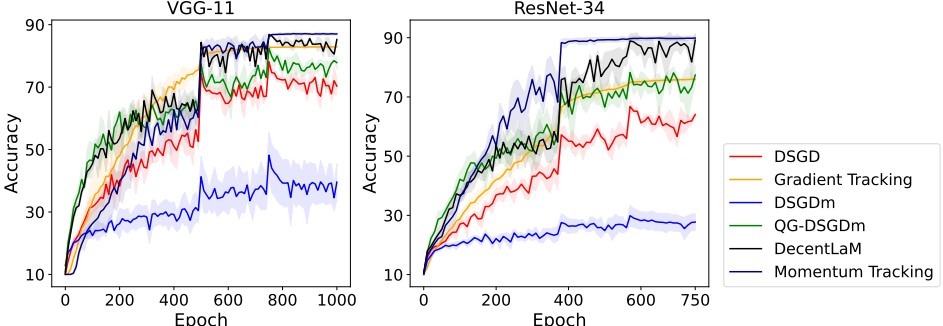

Figure 2: Learning curves for CIFAR-10 with VGG-11 and ResNet-34 in the 2-class setting.

In summary, when the data distributions are homogeneous, DSGDm, QG-DSGDm, DecentLaM, and Momentum Tracking are comparable and outperform DSGD and Gradient Tracking. When the data distributions are heterogeneous, Momentum Tracking is more robust to data heterogeneity than DSGDm, QG-DSGDm, and DecentLaM, and can outperform all comparison methods.

### 4.3 RESULTS WITH VARIOUS NEURAL NETWORK ARCHITECTURES

Next, we evaluated Momentum Tracking in more detail by varying the neural network architecture. Table 3 lists the test accuracy with VGG-11 (Simonyan & Zisserman, 2015) and ResNet-34 (He et al., 2016) when we set $k$ to $\{2, 4, 10\}$, and Fig. 2 shows the learning curves.

For both neural network architectures, Table 3 reveals that when the data distributions are homogeneous (i.e., 10-class), Momentum Tracking is comparable with DSGDm, QG-DSGDm, and DecentLaM and outperforms DSGD and Gradient Tracking. By contrast, when the data distributions are heterogeneous (e.g., 2-class), Table 3 and Fig. 2 reveal that Momentum Tracking outperforms all comparison methods for both neural network architectures. In particular, Fig. 2 indicates that DSGDm, QG-DSGDm, and DecentLaM are unstable and continue to oscillate in the final training phase, whereas Momentum Tracking converges stably. These results are consistent with those of LeNet presented in Table 2. Therefore, the results indicate that Momentum Tracking is more robust to data heterogeneity than DSGDm, QG-DSGDm, and DecentLaM, and can outperform these methods regardless of the neural network architecture.

## 5 CONCLUSION

In this study, we propose Momentum Tracking, which is a method with momentum acceleration whose convergence rate is proven to be independent of data heterogeneity. More specifically, we provide the convergence rate of Momentum Tracking in the standard deep learning setting, in which the objective function is non-convex and the stochastic gradient is used. Our theoretical analysis reveals that the convergence rate of Momentum Tracking is independent of data heterogeneity for any $\beta \in [0, 1)$. Through image classification tasks, we demonstrated that Momentum Tracking can consistently outperform the decentralized learning methods without momentum acceleration regardless of data heterogeneity. Moreover, we showed that Momentum Tracking is more to data heterogeneity than existing decentralized learning methods with momentum acceleration and can consistently outperform these existing methods when the data distributions are heterogeneous.

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
