# OpenReview forum: "Momentum Tracking: Momentum Acceleration for Decentralized Deep Learning on Heterogeneous Data"
_ICLR.cc/2023/Conference — Submitted to ICLR 2023_

### Official Review · Reviewer_hspD · 2022-10-13

**Confidence:** 2
**Correctness:** 3
**Technical Novelty And Significance:** 2
**Empirical Novelty And Significance:** 2
**Recommendation:** 5

**Clarity, Quality, Novelty And Reproducibility:**

In general, I think this paper is clear, and reproducibility can be ensured because the method is not difficult to implement.
However, I do not think this paper is particularly novel, and it seems that this paper only applies the tracking tricks in optimizers to the decentralized training setting.
As for the overall quality, I think it is weakly below the borderline. The ICLR should have more novel or more significant papers.

**Strength And Weaknesses:**

I'm not an expert in this domain. In fact, one of my major research fields is tracking (object tracking). And object tracking almost has nothing to do with momentum tracking. Honestly speaking, I might not have enough skills to assess this paper. However, I hope my comments can help the authors improve their paper. Moreover, it is exciting to me to see if my assessments could align with the professional optimization people because my engineering practice might also benefit the theory design.

Strengths:
1. It seems that this paper has novel components than previous optimizers. And theorems prove that they are robust to data heterogeneity.
2. In general this paper is well written and it is well grounded to the literature. I think all the relevant papers have been referenced and discussed.
3. The experimental results seem promising and somewhat support the authors' claims that this optimizer can augment the decnetralized learning algorithms.

Weaknesses:
1. I do not think this paper is targeting an important enough and the improvements seem incremental. In other words, they do not highlight the core problems in the baselines. Since the data heterogeneity is already well studied. In the introduction, the authors' writing flow seems to be this:
(1) Neural networks are powerful and decentralized learning is studied due to privacy concerns.
(2) DSGDm can work poorly because the parameter drifts away easily.
(3) Gradient tracking literature does not consider the data heterogeneity problem.
However, I do not think the above can lead to the major contribution to this paper and the improvements seem to be A+B (decentralized learning + gradient tracking stuff). It is unclear how important the proposed approach is in the context of decentralized learning literature. Is it a simple borrow from the standard optimization to the decentralized learning setting? If so, how to reflect the novelty and significance of such a borrow? Would it lead to a major change in the results?
Besides, it is unclear to me how important the data heterogeneity problem is and what the key insight for this solution. There is only one sentence in the introduction talking about the technical part of momentum tracking. It seems the technical contribution is weak.
2. The experimental results are not convincing. Can the proposed approach work on ImageNet? Can the proposed optimizer work for object detection and other tasks? I also think the CIFAR10's experimental results are not convincing enough. For the 10-class cases, the training curves almost overlap with the baseline. The 4 class cases are not as important as the 10 class cases in my point of view. In other words, the improvements are not so significant.

**Summary Of The Paper:**

This paper presents a novel momentum-tracking algorithm, whose convergence rate is independent of data heterogeneity. This paper present this algorithm is independent of data heterogeneity for any momentum coefficient. Moreover, experiments find this paper presents a stronger method than previous decentralized learning methods.

**Summary Of The Review:**

Seems to be an incremental application of the momentum tracking technique.

---

> ### Comment · Area_Chair_Dpe5 · 2022-11-11
> **Reminder about novelty**
>
> Novelty does not necessarily mean that the method has to be something completely new. Using existing method in a different way from existing results can also be novel, as long as the proposed method or theory makes sense, such as leading to significant performance improvement, new insights, or new understandings.
>
> Please comment more on whether the authors’s analysis or method show improvement over existing results, whether the improvement makes sense, and whether the improvement leads to new insight or understandings.

---

> ### Comment · Area_Chair_Dpe5 · 2022-11-11
> **Concerns on your comment**
>
> "This paper is restricted and would only attract only a few people in the optimizer + decentralized training domain."
>
> The decentralized optimization is a very important research topic, and the papers on this topic should be discriminated only because they are of fewer people's interests.
>
> I would like to remind again: Please focus more on technical contents of the paper. Whether a paper is accepted or not should be determined by its contribution!

---

> ### Author Response · Authors · 2022-11-14
> **Reply to hspD  (1/2)**
>
> ## Importance of Convergence Analysis
> First, we would like to emphasize that in the decentralized learning literature, **it is required not only to propose a novel algorithm but also to analyze how the convergence rate is affected by the data heterogeneity $\zeta^2$, the number of nodes $N$, and so on.**
> Then, to develop decentralized learning methods that are robust to the data heterogeneity, it is important to prove that their convergence rates are independent of the data heterogeneity $\zeta^2$ [3,4].
>
> ## Data Heterogeneity
> >I do not think this paper is targeting an important enough [...] Since the data heterogeneity is already well studied.
>
> As we mentioned in Sec. 1, the existing studies that addressed the data heterogeneity only considered **the case when the momentum is not applied**.
> In the non-distributed (i.e., non-decentralized) setting, the momentum (i.e., SGDm) becomes an indispensable tool to improve the accuracy of neural networks.
> However, as our experiments indicate, the accuracy of the straightforward method with momentum (i.e., DSGDm) underperforms DSGD (i.e., without momentum) when the data distributions are heterogeneous.
>
> To address this issue, QG-DSGDm (Lin et al., 2021) and DecentLaM (Yuan et al., 2021) have been proposed, but their convergence rates are proven to depend on the data heterogeneity and their accuracy remain to decrease when the data distributions are heterogeneous.
> Therefore, it is important to develop a decentralized learning method with the momentum that is more robust to the data heterogeneity.
> In particular, as we mentioned above, **it is important to propose a method whose convergence rate is proven to be independent of the data heterogeneity $\zeta^2$ even when the momentum is applied** (i.e., Momentum Tracking).
>
> > It is unclear to me how important the data heterogeneity problem is [...]
>
> We would like to emphasize that the data heterogeneity issue is not a new problem setting proposed in our study.
> As we mentioned in Sec. 1, the previous works reported that the data distributions become statistically heterogeneous in the real-world decentralized learning setting [1,2], and it is well-known in the decentralized learning literature that the accuracy of the straightforward methods (e.g., DSGD and GSDGm) decreases when the data distribution is heterogeneous.
> **The importance of addressing data heterogeneity is shared by the decentralized learning literature**.
>
> ## Non-Triviality of Convergence Analysis
> > However, I do not think the above can lead to the major contribution to this paper and the improvements seem to be A+B (decentralized learning + gradient tracking stuff). [...] There is only one sentence in the introduction talking about the technical part of momentum tracking. It seems the technical contribution is weak.
>
> We respectfully disagree with this reviewer's statement.
> Our main contribution is not so much that we proposed a new method called Momentum Tracking, but that we analyzed its convergence rate and proved that it is independent of data heterogeneity.
> **In the setting where the momentum is applied**, the property that the convergence rate is independent of the data heterogeneity has not been obtained in the existing methods and is obtained **for the first time** in Momentum Tracking.
>
> As we stated in Sec. 3.2, we propose Momentum Tracking inspired by Gradient Tracking (Nedic et al., 2017) whose convergence rate has been proven to be independent of the data heterogeneity $\zeta^2$ **when the momentum is not applied**.
> Then, the momentum is a well-known technique to improve the performance of neural networks.
> However, **this does not indicate that it is trivial to analyze the convergence rate of Momentum Tracking**.
>
> Moreover, the experimental results also indicate that Momentum Tracking can outperform the existing decentralized learning method with momentum (i.e., DSGDm, QG-DSGDm, and DecentLaM) when the data distributions are heterogeneous.
> Therefore, Momentum Tracking is theoretically and experimentally more robust to the data heterogeneity than the existing decentralized learning method with momentum.
> For decentralized learning applications, we believe it is important to develop methods that are robust to data heterogeneity while being accelerated by the momentum.
>
> ## Reference
> [1] Hsieh et al., The non-IID data quagmire of decentralized machine learning. In ICML, 2020.
>
> [2] Jean et al., FLamby: Datasets and benchmarks for cross-silo federated learning in realistic healthcare settings, in NeurIPS, 2022.
>
> [3] Tang et al., d2: Decentralized training over decentralized data. In ICML, 2018.
>
> [4] Koloskova el al., An improved analysis of gradient tracking for decentralized machine learning. In NeurIPS, 2021.

---

> > ### Author Response · Authors · 2022-11-14
> > **Reply to hspD (2/2)**
> >
> > ## Experiments
> > >The experimental results are not convincing. [...] Can the proposed optimizer work for object detection and other tasks?
> >
> > The main contribution of our study is the theoretical results (i.e., Theorem 1) (see above reply).
> > In the experiments, we confirmed that Momentum Tracking is more robust to the data heterogeneity than the existing methods through the image classification tasks.
> > Further analysis to evaluate Momentum Tracking in object detection tasks or other tasks is beyond our current scope and can be considered in future work.
> >
> > > The 4 class cases are not as important as the 10 class cases in my point of view.
> >
> > We respectfully disagree with this reviewer's statement.
> > The importance of addressing data heterogeneity is shared by the decentralized learning literature. (See above replay, ``Data Heterogeneity''.)

---

> ### Comment · Area_Chair_Dpe5 · 2022-12-13
> **Notice**
>
> Dear Reviewer,
>
> If you do not want to update your score, please at least acknowledge that you have already read the rebuttal.

---

> ### Comment · Reviewer_hspD · 2022-12-13
> **Acknowledgements**
>
> The reviewer reads the rebuttal and appreciates the authors' effort. However, the reviewer decides to keep the scores unchanged.

---

### Official Review · Reviewer_AZPB · 2022-10-24

**Confidence:** 4
**Clarity, Quality, Novelty And Reproducibility:** Please refer to the previous section.
**Correctness:** 3
**Technical Novelty And Significance:** 2
**Empirical Novelty And Significance:** 2
**Recommendation:** 5

**Strength And Weaknesses:**

Strength
1. The idea of tracking momentum rather than gradient in decentralized learning is interesting, and is more close to real-world applications. In many applications, especially vision tasks, applying momentum is a must.
2. The authors compare both empirical and theoretical results with multiple baselines. Although there are a few missing (details in weakness), but I appreciate the authors for detailed discussion with these mentioned baseline works.

Weakness
1. My main concern is on the claim of "momentum tracking is invariant to the data heterogeneity". In fact, this is not a new result in the decentralized learning. Many algorithms including D2 (https://arxiv.org/pdf/1803.07068.pdf), DeTAG (https://arxiv.org/pdf/2006.08085.pdf), DSGT (https://arxiv.org/pdf/1909.02712.pdf) are able to achieve this with only gradient tracking type methods. The difference is that these existing rates will depend on the $\zeta$ at step 0, which is the norm bound on the $$\zeta_0=\left\|\nabla F_i(x_i^{(0)};\xi_i^{(0)}) - \frac{1}{N}\sum_{j=1}^{N}\nabla F_j(x_j^{(0)};\xi_j^{(0)})\right\|$$.
It seems Theorem 1 in the paper solves this problem with no $\zeta_0$ shown, but it appears that $u_i$ and $c_i$ are initialized to $\zeta_0$. This raises two problems: 1) if $u_i$ is initialized to $0$ (which is usual in practice), it appears the obtained rate is no better than the rates given in the previous works mentioned (D2, DeTAG, DSGT); 2) In practice, it is generally impossible to initialize $u_i$ to such quantity, unless all the nodes all reduce their first gradients.
2. In the experiments, I believe the authors should also compare with the naive baseline where gradient tracking and local momentum are enabled. This way, we can know clearly whether it is tracking or momentum (local) that makes the difference in small-class case.

**Summary Of The Paper:**

This paper proposes momentum tracking in decentralized learning. The authors establish a main theorem that shows the proposed momentum tracking method is invariant to the data heterogeneity bound. The authors conduct several experiments comparing momentum tracking with other baseline methods including gradient tracking (a special case with $\beta=0$), DSGDm, QG-DSGDm and DecentLaM.

**Summary Of The Review:**

I believe the paper has good topic and some interesting analysis, but I'd like to inquire the authors on the main claim they make (momentum tracking is invariant to the data heterogeneity); please refer to the weakness 1 for details.

---

> ### Author Response · Authors · 2022-11-11
> **Reply to AZPB**
>
> ## Novelty on Independence of Data Heterogeneity When Using Momentum
> > My main concern is on the claim of "momentum tracking is invariant to the data heterogeneity". In fact, this is not a new result in the decentralized learning.
>
> We would like to emphasize our contribution and difference from the previous works:
> **in the setting where the momentum is applied**, the property that the convergence rate is independent of the data heterogeneity has not been proven in these existing methods (e.g., $D^2$, DeTAG, and DSGT) and is proven **for the first time** in Momentum Tracking.
>
> These existing studies only considered the case when the momentum is not applied.
> Therefore, it has not been proved whether these methods' convergence rates are independent of the data heterogeneity $\zeta^2$ for any momentum coefficient $\beta \in [0,1)$ or for only the restricted range of $\beta$ (e.g., only when $\beta$ is sufficiently small).
>
> Unlike these studies, we consider the case when the momentum is applied and prove that the convergence rate of Momentum Tracking is independent of the data heterogeneity for any $\beta \in [0,1)$.
> Hence, as mentioned in Secs. 1 and 3.4, Momentum Tracking is **the first method with momentum acceleration** whose convergence rate is proven to be independent of the data heterogeneity in the standard deep learning setting.
>
> ## Initial Values
> > The difference is that these existing rates will depend on the $\zeta_0$ at step 0. [...] if $u_i$ is initialized to 0 (which is usual in practice), it appears the obtained rate is no better than the rates given in the previous works mentioned (D2, DeTAG, DSGT)
>
> We respectfully disagree with this reviewer's statement.
> The difference between our study and these existing studies is that these existing studies (e.g., $D^2$, DeTAG, and DSGT) considered **only the case without momentum**, whereas we consider **the case with momentum** (See above reply).
>
> In the following, we show a more detailed discussion about the initial values of $c_i$ and $u_i$.
> If we initialize $c_i$ and $u_i$ to zeros, Momentum Tracking converges as follows:
> \begin{align}
>     \frac{1}{R} \sum_{r=0}^{R-1} \mathbb{E} \| \nabla f (\bar{x}^{(r)}) \|^2
>     \leq O \left( \sqrt{ \frac{r_0 \sigma^2 L }{N R} }
>     + \left( \frac{r_0^2 L^2 (\tilde{\sigma}^2 + \tilde{\zeta}_0^2)}{p^4 R^2 } ( 1 + \frac{p \beta^2}{1-\beta} ) \right)^{\frac{1}{3}}
>     + \frac{L r_0}{(1-\beta) p^2 R} \sqrt{1 + \frac{\beta^2}{(1-\beta^2)^3 p}} \right),
> \end{align}
> where $\tilde{\sigma}^2 \coloneqq \frac{\sigma^2}{1-\beta}$, $\tilde{\zeta}_0^2 \coloneqq \frac{\zeta_0^2}{p}$, and $\zeta_0^2 \coloneqq \frac{1}{N} \sum_i \| \nabla f_i (\bar{x}^{(0)}) - \nabla f (\bar{x}^{(0)}) \|^2$.
> Therefore, as the reviewer pointed out, if we initialize $c_i$ and $u_i$ to zeros, the convergence rate of Momentum Tracking becomes dependent on $\zeta_0$ but remains to be independent of $\zeta$ as well as $D^2$, DeTAG, and DSGT.
> However, these existing studies only considered the case without momentum.
> It remains to be unclear whether the convergence rates of these existing methods are independent of $\zeta$ when the momentum is applied.
>
> Thus, for both initial values, the advantage of Momentum Tracking over these existing methods is that the convergence rate is proven to be independent of $\zeta^2$ **even when the momentum is applied**.
>
> > In practice, it is generally impossible to initialize $u_i$ to such quantity, unless all the nodes all reduce their first gradients.
>
> As the reviewer mentioned, All-Reduce is necessary to initialize $u_i$ and $c_i$ as in Theorem 1.
> However, as we replied above, the advantage of Momentum Tracking over these previous works (e.g., D2, DeTAG, DSGT) is that the convergence rate of Momentum Tracking is proven to be independent of the data heterogeneity $\zeta^2$ **even when the momentum is applied**.
> This advantage is also valid when we initialize $u_i$ and $c_i$ to zeros.
>
> ## Experimental Results of Naive Baseline
> > The authors should also compare with the naive baseline where gradient tracking and local momentum are enabled.
>
> Thank you for the suggestion.
> We show the results on CIFAR-10 with LeNet.
>
> The results indicate that in the 10-class setting, Gradient Tracking with local momentum outperforms Gradient Tracking and is comparable with Momentum Tracking.
> On the other hand, in the 4-class setting, the accuracy of Gradient Tracking with local momentum decreases by about $29$% compared to the 10-class setting, and Momentum Tracking and Gradient Tracking outperform Gradient Tracking with local momentum.
>
> Thus, Momentum Tracking is more robust to the data heterogeneity than Gradient Tracking with local momentum.
>
> |          | 10-class |  4-class |
> | -------- | -------- | -------- |
> | Gradient Tracking | $62.3 \pm 0.73$ | $61.8 \pm 0.82$ |
> | Gradient Tracking + Local Momentum | $71.0 \pm 0.18$  | $42.2 \pm 4.89$ |
> |Momentum Tracking                   | $\bf{72.9 \pm 0.59}$ | $\bf{70.7 \pm 1.38}$ |

---

> ### Comment · Reviewer_AZPB · 2022-11-14
> **Thanks for the response**
>
> I thank the author for answering my questions. Based on the discussion, the authors and I reach consensus that: the proposed algorithm does depend on $\zeta_0$ just like the baseline algorithms (D2, DeTAG, DSGT). And so it is not entirely invariant to data heterogeneity.
>
> On the other hand, proving gradient tracking with momentum has been analyzed by previous works including Xin & Khan (2020) and Carnevale et al. (2022). Although the authors point out that these baseline results are for strongly convex problems, I'm not sure extending the results to non-convex setting is novel enough because the "tracking" part incurs no explicit challenge for the non-convex problems.
>
> The facts above make this work look a bit incremental. I'd like to question that: (1) is it possible to get rid of the $\zeta_0$ (so that it'd be fundamentally different from previous algorithms); (2) what are the main technical challenges in proving non-convex convergence than strongly convex baseline works?

---

> > ### Author Response · Authors · 2022-11-16
> > **Reply to AZPB**
> >
> > Thank you for the reply.
> >
> > ## Initial Values
> > > Based on the discussion, the authors and I reach consensus that: the proposed algorithm does depend on $\zeta^2_0$ just like the baseline algorithms (D2, DeTAG, DSGT). And so it is not entirely invariant to data heterogeneity. [...] is it possible to get rid of the $\zeta^2_0$ (so that it'd be fundamentally different from previous algorithms)
> >
> > We assume that the reviewer misunderstood that
> > $\frac{1}{1 - \beta} (\nabla F_i (\mathbf{x}_i^{(0)}; \xi_i^{(0)}) - \frac{1}{N} \sum_j \nabla F_j (\mathbf{x}_j^{(0)}; \xi_j^{(0)}))$,
> >  which is shown as the initial values of $\mathbf{u}_i$ and $\mathbf{c}_i$ in Theorem 1, can not be computed in a decentralized manner.
> >
> > One of the simplest methods to compute
> > this initial value is All-Reduce, but as the reviewer mentioned, All-Reduce can not work in a decentralized manner in general.
> > However, several alternative algorithms have been proposed to compute such average values in a fully decentralized manner [1,2].
> > For instance, by using the method shown in Vogels et al. (2022), the average values (e.g., $\frac{1}{N} \sum_{j=1}^N \nabla F_j (\mathbf{x}_j^{(0)}; \xi_j^{(0)})$) can be computed in $D$ steps in a fully decentralized manner, where $D$ is the diameter of the network topology.
> >
> > Therefore, by using this algorithm to initialize $\mathbf{u}_i$ and $\mathbf{c}_i$ as in Theorem 1,
> > **the convergence rate of Momentum Tracking can become independent of $\zeta^2$ and $\zeta^2_0$,
> > and all update procedures can be implemented in a fully decentralized manner**.
> >
> > I would like to summarize our previous and this replies and emphasize the following points:
> > * In both cases where we initialize $\mathbf{u}_i$ and $\mathbf{c}_i$ as in Theorem 1 and where we initialize $\mathbf{u}_i$ and $\mathbf{c}_i$ to zeros, the advantage of Momentum Tracking over the previous studies (e.g., D2 and DSGT) is that we prove the convergence rate of Momentum Tracking is independent of $\zeta^2$ **even when the momentum is applied**.
> > * As we replied above, **the initial values of $\mathbf{u}_i$ and $\mathbf{c}_i$ shown in Theorem 1 can be computed in a fully decentralized manner**. Then, if we initialize $\mathbf{u}_i$ and $\mathbf{c}_i$ as in Theorem 1, the convergence rate of Momentum Tracking is entirely independent of the data heterogeneity (i.e., $\zeta^2$ and $\zeta^2_0$).
> >
> > If we missed the reviewer's concerns, please let us know.
> > We are happy to address your concerns.
> >
> > ## Non-Triviality of Convergence Analysis in Non-Convex Setting
> > >what are the main technical challenges in proving non-convex convergence than strongly convex baseline works?
> >
> > The proof of these previous studies (Xin \& Khan, 2020, Carnevale et al., 2022) is designed with the assumption of using the favorable properties of strong convexity (e.g., Lemma 3  in Xin \& Khan, 2020).
> > However, to derive the convergence rate in the non-convex setting, we can not use these properties,
> > and the essential inequalities used in their proofs do not hold in the non-convex setting,
> > which is the main difficulty in extending their proof techniques in the non-convex setting.
> >
> > Moreover, roughly speaking, Xin \& Khan, (2020) and Carnevale et al., (2022) provided the upper bound of $\| \bar{\mathbf{x}}^{(r)} - \mathbf{x}^\star \|$ as the convergence rate where $\mathbf{x}^\star$ is the global solution of decentralized learning.
> > However, to provide the convergence rate in the non-convex setting, we need to provide the upper bound of $\| \nabla f (\bar{\mathbf{x}}^{(r)}) \|$,
> > which makes it more challenging to use their proof techniques in the non-convex setting.
> >
> > Thus, it was difficult to extend these proofs (Xin \& Khan, 2020, Carnevale et al., 2022) to the non-convex setting.
> > Our proof is based on a different study (Koloskova et al. 2020) that analyzed the convergence rate of DSGD in the non-convex and stochastic setting,
> > and we show that the data heterogeneity $\zeta^2$ can be eliminated from the convergence rate by Momentum Tracking.
> > See the reply to uTVi, "Proof Sketch," for a more detailed explanation of how we eliminate the dependence of the data heterogeneity $\zeta^2$ from the convergence rate,
> > which are the most important component of our proof.
> >
> > ## Reference
> >
> > [1] Chih-Kai Ko. On Matrix Factorization and Scheduling for Finite-time Average-consensus. PhD thesis, 2010.
> >
> > [2] Thijs Vogels et al., RelaySum for decentralized deep learning on heterogeneous data. In NeurIPS, 2021.

---

### Official Review · Reviewer_uTVi · 2022-10-24

**Confidence:** 5
**Correctness:** 3
**Technical Novelty And Significance:** 2
**Empirical Novelty And Significance:** 3
**Recommendation:** 5

**Clarity, Quality, Novelty And Reproducibility:**

Clarity: It is not clear how the proposed algorithm relates to gradient tracking.

Novelty: The idea of using a correction term to address the distribution shift issue has been extensively studied in FL. It would be good to provide more discussions.

Reproducibility: Good.

**Strength And Weaknesses:**

Pros:
1. The problem studied is important and the proposed algorithm looks effective based on the empirical results.
2. The authors provided extensive experimental results.

Cons:
1. The authors claim that the proposed algorithm becomes gradient tracking when $\beta=0$. However, it is not clear to me. In particular, in gradient tracking, there should be a difference between the stochastic gradient in two consecutive iterations on each device. But there is no such a term.
2. Introducing a correction term to the local gradient is a commonly used approach to address the distribution shift issue. For instance, SCAFFOLD also employs such a strategy. What is the difference between your approach and SCAFFOLD?
3. As for the convergence rate, existing algorithms only have a quadratic dependence on the spectral gap. But the proposed algorithm has a worse dependence. Is it possible to improve this dependence? If not, what is the reason? More discussions are needed.


**Summary Of The Paper:**

This paper developed a new decentralized optimization algorithm to address the heterogeneous data distribution issue. In particular, it introduces a correction term for the local stochastic gradient to alleviate the distribution shift issue. The authors further provided theoretical analysis for the convergence rate and conducted empirical results to demonstrate the performance of the proposed algorithm.



**Summary Of The Review:**

Overall, this paper developed a new decentralized optimization approach. But some parts are not quite clear, the theoretical bound is not tight, and the novelty is incremental.

---

> ### Author Response · Authors · 2022-11-10
> **Reply to uTVi**
>
> ## Relationship with Gradient Tracking
> > The authors claim that the proposed algorithm becomes gradient tracking when $\beta=0$. However, it is not clear to me.
>
> We assume that the reviewer misunderstood that Eqs. (7-9) are not equivalent to the update rules of Gradient Tracking when $\beta=0$.
> Here, we prove that Momentum Tracking becomes Gradient Tracking when $\beta=0$.
>
> When $\beta=0$, the update rules of Momentum Tracking are obtained as follows:
> $$ x_i^{(r+1)} = \sum_{j \in N_i^+} W_{ij} x_j^{(r)} - \eta ( \nabla F_i (x_i^{(r)} ; \xi_i^{(r)}) - c_i^{(r)} ) $$
> $$ c_i^{(r+1)} = \sum_{j \in N_i^{+}} W_{ij}  ( c_j^{(r)} - \nabla F_j(x_j^{(r)} ; \xi_j^{(r)}) ) + \nabla F_i(x_i^{(r)} ; \xi_i^{(r)}) $$
> Then, we define $y_i^{(r)} \coloneqq \nabla F_i( x_i^{(r)} ; \xi_i^{(r)}) - c_i^{(r)}$ and replace $c_i$ with $y_i$.
> We then obtain the following update rules:
> $$  x_i^{(r+1)} = \sum_{j \in N_i^+} W_{ij} x_j^{(r)} - \eta y_i^{(r)} $$
> $$ y_i^{(r+1)} = \sum_{j \in N_i^{+}} W_{ij} y_j^{(r)} + \nabla F_i ( x_i^{(r+1)} ; \xi_i^{(r+1)}) - \nabla F_i( x_i^{(r)} ; \xi_i^{(r)}) $$
> Because the reformulated update rules are equivalent to that of DIGing (Nedic et. al. 2017), which is one of the most famous variants of Gradient Tracking, Momentum Tracking becomes Gradient Tracking when $\beta=0$.
>
> If we miss some of your points, please let us know.
> We are more than happy to address your concerns.
>
> ## Novelty on Independence of Data Heterogeneity When Using Momentum
> > Introducing a correction term to the local gradient is a commonly used approach to address the distribution shift issue. [...] What is the difference between your approach and SCAFFOLD?
>
> The main difference between Momentum Tracking and SCAFFOLD is that the convergence rate of SCAFFOLD has been proven to be independent of the data heterogeneity **only when the momentum is not applied**,
> whereas the convergence rate of Momentum Tracking is proven to be independent of the data heterogeneity **when the momentum is applied**.
>
> As the reviewer mentioned, there are many methods, including SCAFFOLD, $D^2$, and Gradient Tracking, that addresses the data heterogeneity by using the tracking mechanism.
> However, these existing studies did not prove that convergence rates are independent of the data heterogeneity when the momentum is applied.
> Thus, when we use the momentum, it remains to be unclear whether these tracking mechanisms can make the methods independent of the data heterogeneity or whether the additional technique is necessary.
> Unlike these existing studies, we consider the case when the momentum is applied.
> Then, we propose Momentum Tracking, which is **the first method with momentum acceleration** whose convergence rate is proven to be independent of the data heterogeneity.
>
> ## Discussion on Deriving Better Convergence Rate
> > As for the convergence rate, existing algorithms only have a quadratic dependence on the spectral gap [..] Is it possible to improve this dependence? If not, what is the reason?
>
> We could improve the current convergence rate of Momentum Tracking with $\beta=0$ by using some techniques proposed by Koloskova et al. (2021) in terms of the spectral gap.
> Moreover, by using this proof technique, we may improve the convergence rate of Momentum Tracking when $\beta > 0$. (See Sec. B.)
>
> Nevertheless, our main contribution is providing the convergence rate of Momentum Tracking that is independent of the data heterogeneity for any momentum coefficient $\beta \in [0,1)$.
> Further analysis to improve other variables is beyond our current scope and can be considered in future work.

---

> > ### Comment · Reviewer_uTVi · 2022-11-11
> > **More questions on Independence of Data Heterogeneity**
> >
> > Thanks for the authors' response.
> >
> > I have more questions about the Independence of Data Heterogeneity.
> >
> > Which steps or what strategies in your proof help you remove the dependence of data heterogeneity?
> >
> > As you know, when bounding the consensus error, it is very likely that the heterogeneity hyperparameter $\zeta^2$ is introduced into the error bound. How does your proof avoid this term? Which step can introduce $\zeta^2$  in the convergence rate if not using momentum tracking?
> >
> > It would be good if the authors could provide more discussion about the uniqueness of the proof.

---

> > > ### Author Response · Authors · 2022-11-14
> > > **Proof Sketch (3/3)**
> > >
> > > > It would be good if the authors could provide more discussion about the uniqueness of the proof.
> > >
> > > Our proof is inspired by the analysis of DSGD (Koloskova et al. 2020).
> > > However, the convergence rate of DSGD depends on the data heterogeneity $\zeta^2$.
> > > To the best of our knowledge, the proof strategy to derive the upper bound of $\Xi$ that is independent of $\zeta^2$ by combining the three inequalities about $\Xi$, $\mathcal{E}$, and $\mathcal{D}$, which we explained above, is novel.
> > >
> > > In particular, due to the momentum, $\mathcal{D}$ appears in the inequality of $\mathcal{E}$, and two inequalities about $\mathcal{E}$ and $\mathcal{D}$ become necessary,
> > > which makes the convergence analysis difficult.
> > > Moreover, to drive the convergence rate that holds for any momentum coefficient $\beta \in [0,1)$, it is required to tune the coefficient of the inequalities carefully (e.g., $\frac{2 \beta^2}{1 + \beta^2} (< 1)$ in the above inequality of $\mathcal{D}$, and the definitions of $A$ in $\Theta$ and $t$).
> > >
> > > Thus, it is not trivial to prove the convergence rate of Momentum Tracking is independent of $\zeta^2$, and in particular for any $\beta \in [0,1)$.

---

> > > ### Author Response · Authors · 2022-11-14
> > > **Proof Sketch (2/3)**
> > >
> > > **Proof Sketch:**
> > > We derive the inequality about the consensus error $\Xi$ as follows (See Lemma 14):
> > > \begin{align}
> > >     \Xi^{(r+1)}
> > >     \leq \left(1 - \frac{p}{2}\right) \Xi^{(r)}
> > >     + \frac{9}{p} \eta^2 \mathcal{E}^{(r)}
> > >     + \frac{9}{N p} \eta^2 \sum_{i=1}^{N} \left( \mathbb{E} \left\|  \mathbf{u}_i^{(r+1)} - \mathbf{d}_i^{(r+1)} \right\|^2
> > >     +  \mathbb{E} \left\| \mathbf{e}_i^{(r+1)} \right\|^2 \right)
> > > \end{align}
> > >
> > >
> > > $\mathcal{E}$ represents the the error between global momentum $\mathbf{e}_i (= \bar{\mathbf{e}})$ and corrected local momentum $(\mathbf{d}_i - \mathbf{c}_i$).
> > > Thus, intuitively, if we remove the tracking term $\mathbf{c}_i$ from Momentum Tracking,
> > > $\mathcal{E}$ becomes $\frac{1}{N} \sum_i \mathbb{E} \| \mathbf{d}_i  - \mathbf{e}_i \|^2$, which causes the data heterogeneity $\zeta^2$ to appear in the upper bound of $\Xi$.
> > > In the following, we explain how to eliminate $\mathcal{E}^{(r)}$ and $\zeta^2$ from the upper bound of $\Xi^{(r+1)}$,
> > > which is the most important component of our proof.
> > >
> > > To bound $\mathcal{E}$ from above, we derive the following two inequalities
> > > (see Lemmas 15 and 16):
> > > \begin{align}
> > >     \mathcal{E}^{(r+1)}
> > >     &\leq \left(1 - \frac{p}{2}\right) \mathcal{E}^{(r)}
> > >     + \frac{18 \beta^2}{p} \mathcal{D}^{(r)}
> > >     + \frac{144 L^4}{p} \eta^2 \Xi^{(r)}
> > >     + C_1 \\\\
> > >     \mathcal{D}^{(r+1)}
> > >     &\leq \frac{2 \beta^2}{1+\beta^2} \mathcal{D}^{(r)}
> > >     + \frac{32 L^4 \eta^2}{1 - \beta^2} \Xi^{(r)}
> > >     + C_2,
> > > \end{align}
> > > where $C_1$ and $C_2$ are variables independent of $\Xi$, $\mathcal{D}$, $\mathcal{E}$, and $\zeta^2$.
> > > Here, the most important benefit to adding the tracking term $\mathbf{c}_i$ is that the coefficient of $\mathcal{E}^{(r)}$ becomes less than 1. (i.e., $(1 - \frac{p}{2}) < 1$).
> > > Roughly speaking, the above two inequalities imply that $\mathcal{E}$ and $\mathcal{D}$ become gradually smaller
> > > because $(1 - \frac{p}{2}) < 1$ and $\frac{2 \beta^2}{1 + \beta^2} < 1$ hold for any $\beta \in [0,1)$.
> > >
> > > Next, we derive a new inequality by combining the above three inequalities.
> > > We define an auxiliary error term $\Theta$ as follows:
> > > \begin{align}
> > >     \Theta^{(r)} \coloneqq \Xi^{(r)} +  \frac{36}{p^2} \eta^2 \mathcal{E}^{(r)} + \frac{A \beta^2}{p^3} \eta^2 \mathcal{D}^{(r)},
> > > \end{align}
> > > where $A > 0$ is defined in Lemma 17.
> > > Then, by combining the above three inequalities, we obtain the following (see Lemma 17):
> > > \begin{align}
> > >     \Theta^{(r+1)}
> > >     &\leq \left(1 - \frac{p}{t}\right) \Theta^{(r)}
> > >     + C_3,
> > > \end{align}
> > > where $C_3$ is a variable independent of $\Xi$, $\mathcal{D}$, $\mathcal{E}$ and $\zeta^2$, and $t \geq 4$ is defined in Lemma 17.
> > >
> > > Using $\Xi \leq \Theta$ and applying the above inequality recursively, we obtain (see Lemma 18)
> > > \begin{align}
> > >     \Xi^{(r+1)} \leq \Theta^{(r+1)}
> > >     \leq\left(1 - \frac{p}{t}\right)^{r+1} \Theta^{(0)}
> > >     + C_4,
> > > \end{align}
> > > where $C_4$ is a variable independent of $\Xi$, $\mathcal{D}$, $\mathcal{E}$, and $\zeta^2$.
> > >
> > > Using $(1 - \frac{p}{t}) < 1$, it holds that $\sum_{r=0}^R (1 - \frac{p}{t})^{r+1} \Theta^{(0)} \leq \frac{t}{p} \Theta^{(0)}$.
> > > Finally, using this inequality and the fact that $C_4$ and $\Theta^{(0)}$ are independent of $\zeta^2$ due to the assumption of initial values,
> > > we can eliminate $\mathcal{E}$ from the upper bound of $\Xi$ and derive the upper bound of $\Xi$ that does not contain $\zeta^2$ as follows (see Lemma 19):
> > > \begin{align}
> > >     \frac{4 L^2}{R+1} \sum_{r=0}^{R} \Xi^{(r)}
> > >     &\leq \frac{1}{2 (R+1)} \sum_{k=0}^{R} \left\| \nabla f(\bar{\mathbf{x}}^{(k)}) \right\|^2
> > >     + \frac{40 L^2 t}{(1-\beta)^3 p^2} \left( 10 + \frac{29}{p} + \frac{864}{p^2} \right) \sigma^2 \eta^2.
> > > \end{align}
> > >
> > > These are the essential techniques to bound $\sum_{r=0}^R \Xi^{(r)}$ from above without using $\zeta^2$ and make the convergence rate independent of $\zeta^2$.
> > > If the reviewer has questions about this proof sketch, we are happy to address them.
> > >
> > > > Which step can introduce $\zeta^2$ in the convergence rate if not using momentum tracking?
> > >
> > > If we omit the tracking term $\mathbf{c}_i$ from Momentum Tracking,
> > > $\mathcal{E}^{(r)}$ in the above inequality of $\Xi$ becomes to introduce $\zeta^2$ (see above reply).

---

> > > ### Author Response · Authors · 2022-11-14
> > > **Proof Sketch (1/3)**
> > >
> > > Thank you for the reply.
> > >
> > > > Which steps or what strategies in your proof help you remove the dependence of data heterogeneity? [...] when bounding the consensus error, it is very likely that the heterogeneity hyperparameter $\zeta^2$ is introduced into the error bound. How does your proof avoid this term?
> > >
> > > **TL;DR:**
> > > First, we briefly summarize our proof technique.
> > > Our proof is based on the analysis of DSGD (Koloskova et al. (2020)) that uses the upper bound of the consensus error.
> > > We extend their technique to deal with the heterogeneity $\zeta^2$ by decomposing the inequality about the consensus error into
> > > (i) the inequality of the error between global and **corrected** local momentum
> > > and (ii) the inequality of the error between update values of global and **uncorrected** local momentum.
> > > While bounding the latter error is rather straightforward, bounding the former error requires our momentum correction term $\mathbf{c}_i$, which makes the error recursion contractive;
> > > otherwise, the error between global and local momentum results in the heterogeneity term in the final convergence error.
> > >
> > > In the following, we show the proof sketch and explain in more detail how to bound the consensus error from above without using $\zeta^2$.
> > >
> > > **Notation:**
> > > We define the update rules of auxiliary variables $\mathbf{d}_i$ and $\mathbf{e}_i$ as follows:
> > > \begin{align}
> > >     \mathbf{d}_i^{(r+1)} &= \beta \mathbf{d}_i^{(r)} + \nabla f_i(\bar{\mathbf{x}}^{(r)}), && \text{(\emph{uncorrected} local momentum)} \\\\
> > >     \mathbf{e}_i^{(r+1)} &= \beta \mathbf{e}_i^{(r)} + \nabla f(\bar{\mathbf{x}}^{(r)}), && \text{(global momentum)}
> > > \end{align}
> > > where $\mathbf{d}_i^{(0)} = \frac{1}{1-\beta} (\nabla f_i (\bar{\mathbf{x}}^{(0)}) - \nabla f (\bar{\mathbf{x}}^{(0)}))$
> > > and $\mathbf{e}_i^{(0)} = \mathbf{0}$.
> > >
> > > We define $\Xi$, $\mathcal{E}$, and $\mathcal{D}$ as follows:
> > > \begin{align}
> > >     \Xi^{(r)} &\coloneqq \frac{1}{N} \sum_{i=1}^N \mathbb{E} \left\| \mathbf{x}_i^{(r)} - \bar{\mathbf{x}}^{(r)} \right\|^2, && \text{(consensus error)}
> > > \end{align}
> > >
> > > \begin{align}
> > >     \mathcal{E}^{(r)} &\coloneqq \frac{1}{N} \sum_{i=1}^N \mathbb{E} \left\| \mathbf{d}_i^{(r+1)} - \mathbf{c}_i^{(r)} - \mathbf{e}_i^{(r+1)} \right\|^2, && \text{(error between global/\emph{corrected} local momentum)}
> > > \end{align}
> > >
> > > \begin{align}
> > >     \mathcal{D}^{(r)} &\coloneqq \frac{1}{N} \sum_{i=1}^N \mathbb{E} \left\| \left( \mathbf{d}_i^{(r+1)} - \mathbf{d}_i^{(r)} \right) - \left( \mathbf{e}_i^{(r+1)} - \mathbf{e}_i^{(r)} \right) \right\|^2. && \text{(error between update values of global/\emph{uncorrected} local momentum)}
> > > \end{align}

---

> ### Comment · Area_Chair_Dpe5 · 2022-12-13
> **Notice**
>
> Dear Reviewer,
>
> If you do not want to update your score, please at least acknowledge that you have already read the rebuttal.

---

> > ### Comment · Reviewer_uTVi · 2022-12-13
> > **Thanks for the authors' response**
> >
> > Thanks for the authors' response.
> >
> > I have read the authors' responses to my questions and the comments from other reviewers.
> >
> > I agree with Reviewer AZPB. This algorithm is NOT independent of heterogeneity.
> >
> > In particular, the problem does not lie in how to compute $\frac{1}{N} \sum_j \nabla F_j\left(\mathbf{x}_j^{(0)} ; \xi_j^{(0)}\right)$. Instead, you cannot guarantee $\nabla F_i\left(\mathbf{x}_i^{(0)} ; \xi_i^{(0)}\right)-\frac{1}{N} \sum_j \nabla F_j\left(\mathbf{x}_j^{(0)} ; \xi_j^{(0)}\right)=0$ due to the heterogeneous data distribution, i.e., Lemma 4 is NOT correct.
> >
> > Considering this fatal error, I have to lower my rating.

---

> > > ### Author Response · Authors · 2022-12-13
> > > **Response to uTVi**
> > >
> > > > This algorithm is NOT independent of heterogeneity.
> > >
> > > We respectfully disagree with this statement.
> > > We assume that the reviewer misunderstands that the initial values of $c_i$ and $u_i$ in Theorem 1 can not be computed in practice.
> > > However, as replied to Reviewer AZPB, **the initial values of $c_i$ and $u_i$ in Theorem 1 can be computed in a fully decentralized manner** (See [1] and the response to AZPB).
> > > Therefore, the convergence rate of Momentum Tracking is independent of the data heterogeneity $\zeta$ (and $\zeta_0$).
> > >
> > > To address the reviewer's concerns, we promise to add an illustration of the algorithm to compute the initial values of $c_i$ and $u_i$ in a fully decentralized manner, which is proposed in [1], in the revised manuscript.
> > >
> > > Moreover, we would like to note that the convergence rates of D2 and Gradient Tracking, which are claimed to be independent of the data heterogeneity, have been shown to depend on $\zeta_0$ [2,3].
> > >
> > > > Lemma 4 is NOT correct.
> > >
> > > We disagree with this statement.
> > > **Lemma 4 is correct.**
> > > Moreover, Lemma 4 holds in both cases where $c_i$ and $u_i$ are initialized to zeros and where $c_i$ and $u_i$ are initialized as in Theorem 1.
> > >
> > > We assume that the reviewer may misunderstand the final equality of Lemma 4.
> > > Here, we show a more detailed explanation.
> > > When $c_i$ and $u_i$ are initialized to zeros, the final equality of Lemma 4 holds trivially (i.e., $\sum_i c_i^{(0)} = 0$).
> > > When $c_i$ and $u_i$ are initialized as in Theorem 1, we have
> > >
> > > \begin{align*}
> > >     \sum_{i=1}^N c_i^{(0)}
> > >     = \frac{1}{1-\beta} \sum_{i=1}^N \left( \nabla F_i (x_i^{(0)} ; \xi_i^{(0)}) - \frac{1}{N} \sum_{j=1}^N \nabla F_j (x_j^{(0)} ; \xi_j^{(0)}) \right)
> > >     = \frac{1}{1-\beta} \left[ \left( \sum_{i=1}^N \nabla F_i (x_i^{(0)} ; \xi_i^{(0)}) \right) - \left( \sum_{j=1}^N \nabla F_j (x_j^{(0)} ; \xi_j^{(0)}) \right) \right]
> > >     = \mathbf{0}
> > > \end{align*}
> > >
> > > Therefore, Lemma 4 holds in both cases where $c_i$ and $u_i$ are initialized to zeros and where $c_i$ and $u_i$ are initialized as in Theorem 1.
> > >
> > >
> > >  ## Reference
> > > [1] Thijs Vogels et al., RelaySum for decentralized deep learning on heterogeneous data. In NeurIPS, 2021.
> > >
> > > [2] Koloskova et al., An improved analysis of gradient tracking for decentralized machine learning. In NeurIPS, 2021.
> > >
> > > [3] Tang et al., d2: Decentralized training over decentralized data. In ICML, 2018.

---

> > > > ### Comment · Reviewer_uTVi · 2022-12-13
> > > > **Thanks for your clarification**
> > > >
> > > > Lemma 4 is correct. You are right.
> > > >
> > > > But there might be a problem for $\mathcal{E}^{(0)}$ in Page 31. How did you get the second step? It seems you missed a coefficient $\beta$ for the full gradient.
> > > >
> > > > Please correct me if I am wrong.

---

> > > > > ### Author Response · Authors · 2022-12-13
> > > > > **Thank you for the response**
> > > > >
> > > > > > But there might be a problem for $\mathcal{E}^{(0)}$ in Page 31. How did you get the second step? It seems you missed a coefficient $\beta$ for the full gradient.
> > > > >
> > > > > We show a more detailed equation expansion below.
> > > > > From the definition of $\mathcal{E}$, we have
> > > > > \begin{align*}
> > > > >     \mathcal{E}^{(0)}
> > > > >     = \frac{1}{N} \mathbb{E} \left\| \mathbf{D}^{(1)} - \mathbf{C}^{(0)} - \mathbf{E}^{(1)} \right\|^2_F
> > > > > \end{align*}
> > > > > Using the update rules and initial values of $\mathbf{D}$ and $\mathbf{E}$ (i.e., Eqs. (21-22)), we have
> > > > > \begin{align}
> > > > >     \mathbf{D}^{(1)} &= \frac{\beta}{1-\beta} (\nabla f (\bar{\mathbf{X}}^{(0)}) - \frac{1}{N} \nabla f (\bar{\mathbf{X}}^{(0)}) \mathbf{1} \mathbf{1}^\top ) + \nabla f(\bar{\mathbf{X}}^{(0)}), \\\\
> > > > >     \mathbf{E}^{(1)} &= \frac{1}{N} \nabla f(\bar{\mathbf{X}}^{(0)}) \mathbf{1}\mathbf{1}^\top.
> > > > > \end{align}
> > > > > Note that $\mathbf{D}$ and $\mathbf{E}$  are auxiliary variables only used in the proof.
> > > > >
> > > > > Then, we get
> > > > > \begin{align}
> > > > >     \mathbf{D}^{(1)} - \mathbf{E}^{(1)} = \frac{1}{1-\beta} (\nabla f (\bar{\mathbf{X}}^{(0)}) - \frac{1}{N} \nabla f (\bar{\mathbf{X}}^{(0)}) \mathbf{1} \mathbf{1}^\top ).
> > > > > \end{align}
> > > > >
> > > > >
> > > > > Finally, by using the initial values of $c_i$, we get
> > > > > \begin{align*}
> > > > >     \mathcal{E}^{(0)}
> > > > >     = \frac{1}{(1-\beta)^2} \frac{1}{N} \mathbb{E} \left\| \nabla f (\bar{\mathbf{X}}^{(0)}) - \frac{1}{N} \nabla f (\bar{\mathbf{X}}^{(0)}) \mathbf{1} \mathbf{1}^\top - \nabla F (\mathbf{X}^{(0)} ; \xi^{(0)}) + \frac{1}{N} \nabla F (\mathbf{X}^{(0)} ; \xi^{(0)}) \mathbf{1}\mathbf{1}^\top \right\|^2_F.
> > > > > \end{align*}
> > > > > Therefore, the above equation (i.e., the second step of $\mathcal{E}^{(0)}$) is correct.
> > > > >
> > > > > If the reviewer still has concerns about the above equation, we would be happy to address them.

---

> > > > > ### Author Response · Authors · 2022-12-13
> > > > > **Could our response have solved the previous concern?**
> > > > >
> > > > > In the previous response of the reviewer, we think that the main concern of the reviewer is whether the initial values of $u_i$ and $c_i$ in Theorem 1 can be computed or not in practice.
> > > > >
> > > > > Could our previous response have solved this concern?
> > > > > If our previous response did not solve this concern, we are happy to address it.

---

### Official Review · Reviewer_i18n · 2022-10-24

**Confidence:** 4
**Clarity, Quality, Novelty And Reproducibility:** Please check the comments above.
**Correctness:** 3
**Technical Novelty And Significance:** 3
**Empirical Novelty And Significance:** 2
**Recommendation:** 6

**Strength And Weaknesses:**

## Strength
* The paper is very clear and well-written.
* The attempts introduced in the manuscript may alleviate the issue of data heterogeneity; some empirical results also justify its potential.

## Weaknesses
1. Missing comparison with RelaySum (Vogels et al 2021). The reviewer agrees with the authors that the original paper of RelaySum did not provide the convergence rate for DSGDm; however, all their empirical evaluations are on top of DSGDm. As a paper proposed to address the issue of decentralized deep learning, it is equally important to justify the performance gain of the proposed methods over all existing SOTA methods, in terms of their empirical performance.
2. The manuscript considers an uncommon experimental setting for data heterogeneity, unlike the recent standard Dirichlet distributions in the community of decentralized deep learning and federated learning. The authors are required to justify their design choice and at least include some experiments for CIFAR, in terms of alpha=10, 1, 0.1. The authors can also take the chance to evaluate other datasets with more classes and consider other neural architectures (e.g. Vit), as the current evaluations are quite limited.
3. The hyper-parameter settings stated in Appendix E seem a bit random to the reviewer and thus make results less convincing. For example, why 1000 epochs and 750 epochs are used for VGG-11 and ResNet-34, with a relatively small initial learning rate like 0.05? The reviewer also checked the prior work like QG-DSGDm and RelaySum, where in their experiments all settings were well-tuned for each baseline, making the comparison fair enough.
4. Limited topology scale. Only 8 workers are considered for the evaluation, and it is unclear to the reviewer if some issues will occur when scaling up.

**Summary Of The Paper:**

The manuscript tries to address the issue of decentralized deep learning with heterogeneous local data, by extending the idea of gradient tracking to the momentum buffer.
The convergence rate of the proposed idea has no dependence on data heterogeneity and can achieve better empirical performance than existing methods.

**Summary Of The Review:**

The reviewer enjoys reading the manuscript and believes it is a crucial question for decentralized deep learning.

The proposed solution may be a valid method to alleviate the data heterogeneity. However, the current manuscript still needs to improve its experimental settings and compare it fairly with STOA methods.

The reviewer will reconsider the rating once the issues pointed out in the weakness part have been addressed.


### post-rebuttal
still a borderline paper (though some authors' responses have addressed the reviewer's concern, and have raised the score to 6).

---

> ### Author Response · Authors · 2022-11-14
> **Reply to i18n**
>
> ## Comparison with RelaySum
> > Missing comparison with RelaySum (Vogels et al 2021). [...]
>
> Thank you for the suggestion.
> As the reviewer mentioned, the convergence rate of RelaySum has not been proven whether it is independent of the data heterogeneity $\zeta^2$ when the momentum is applied, but Vogels et al. 2021 showed that RelaySum is experimentally robust to the data heterogeneity even when the momentum is applied.
> However, our main contribution is that we propose the method with momentum whose convergence rate is proven to be independent of the data heterogeneity (i.e., Momentum Tracking),
> and it is the clear advantage of Momentum Tracking over RelaySum.
> Therefore, it is beyond our current scope that we compare RelaySum and Momentum Tracking experimentally and then discuss which is experimentally more robust to the data heterogeneity.
>
> ## Additional Experiments on Data Heterogeneity
> >The manuscript considers an uncommon experimental setting for data heterogeneity, unlike the recent standard Dirichlet distributions in the community of decentralized deep learning and federated learning. The authors are required to justify their design choice and at least include some experiments for CIFAR, in terms of alpha=10, 1, 0.1
>
> The $k$-class setting that we used in our experiments follows these previous works [1,2].
> As shown in these studies, we can change the data heterogeneity by using this $k$-class setting as well as by using the setting with Dirichlet distributions.
>
> In addition, we conducted the experiment in the setting where the data are distributed into nodes by using Dirichlet distribution.
> The following table shows the results for CIFAR-10 with VGG when we set $\alpha = 0.1$ (i.e., the heterogeneous setting).
>
> | CIFAR-10 + VGG |  $\alpha = 0.1$ |
> | --------          | -------- |
> | QG-DSGDm          | $87.2 \pm 1.54$ |
> | DecentLaM         | $88.6 \pm 0.99$ |
> | Momentum Tracking | $\bf{90.2 \pm 0.37}$ |
>
>
> The result indicates that Momentum Tracking can outperform QG-DSGDm and DecentLaM when the data distributions are heterogeneous,
> which is consistent with the results in the $k$-class setting.
>
> >The authors can also take the chance to evaluate other datasets with more classes and consider other neural architectures (e.g. Vit), as the current evaluations are quite limited.
>
> Thank you for the suggestion.
> We are now conducting experiments with CIFAR-100 and will add the results in the revised manuscript.
>
> ## Additional Experiments with Larger Learning Rate
> > The hyper-parameter settings stated in Appendix E seem a bit random to the reviewer [...] For example, why 1000 epochs and 750 epochs are used for VGG-11 and ResNet-34, with a relatively small initial learning rate like 0.05?
>
> Because different neural network architectures have different convergence speeds, we used the different total number of epochs for VGG and ResNet. Then, since all methods are compared with the same total number of epochs for each neural network architecture and dataset, all methods are fairly evaluated in our experimental settings.
>
> Thank you for the suggestion about the learning rate.
> In the $2$-class (i.e., heterogeneous) setting, we conducted the additional experiments for CIFAR-10 with VGG and ResNet when we set the learning rate $\eta$ to $\\{0.5, 0.1, 0.05, 0.01, 0.005 \\}$ and compared the results of QG-DSGDm, DecentLaM, and Momentum Tracking.
> The following table shows the results, where the modified result is underlined.
>
> | $2$-class |  CIFAR-10 + VGG | CIFAR-10 + ResNet |
> | -------- | -------- | -------- |
> | QG-DSGDm          | $77.8 \pm 1.96$       |  $\underline{78.2 \pm 2.09 \\;\\; ( + 0.8)}$ |
> | DecentLaM         | $85.2 \pm 0.67$       |  $89.2 \pm 2.26$ |
> | Momentum Tracking | $\bf{ 87.0 \pm 0.48}$ |  $\bf{89.9 \pm 0.73}$ |
>
>
> The results indicate that the accuracy of QG-DSGDm is slightly improved, but Momentum Tracking remains to outperform QG-DSGDm and DecentLaM.
> Therefore, our experimental results that Momentum Tracking is more robust to the data heterogeneity than QG-DSGDm and DecentLaM did not change.
>
> ## Comparison on Large Topology
> >Limited topology scale. Only 8 workers are considered for the evaluation, and it is unclear to the reviewer if some issues will occur when scaling up.
>
> In the experiments shown in Sec. C.4 in the supplementary material,
> we compared the methods in the setting where the underlying network topology is a ring consisting of 25 nodes using the synthetic dataset.
> The results indicate that Momentum Tracking can converge faster than the other comparison methods (e.g., QG-DSGDm and DecentLaM) even when the underlying network topology is relatively large.
>
> ## Reference
> [1] Kenta Niwa, Noboru Harada, Guoqiang Zhang, and W. Bastiaan Kleijn. Edge-consensus learning:
> Deep learning on p2p networks with nonhomogeneous data. In KDD, 2020.
>
> [2] Tehrim Yoon, Sumin Shin, Sung Ju Hwang, and Eunho Yang. Fedmix: Approximation of mixup under mean augmented
> federated learning. In ICLR, 2021.

---

> ### Comment · Reviewer_i18n · 2022-11-16
> **additional review**
>
> The reviewer thanks the authors for providing the feedback.
>
> As somehow has been pointed out in other reviews, the reviewer still believes that the main contribution of the current manuscript is (1) a proper way to integrate gradient tracking with momentum and (2) empirical deep learning performance gains. The proof techniques in Koloskova et al. 2020 are quite standard (as shown in many recent works) and can be easily extended to eliminate data heterogeneity $\zeta$ using some extra tricks.
> * For example, the key step of the proof sketch stated in the [response](https://openreview.net/forum?id=3eQEil044E&noteId=PvSri4Gb1s), i.e., *the idea of decomposing the consensus error* into (i) the inequality of the error between global and corrected local momentum and (ii) the inequality of the error between update values of global and uncorrected local momentum. Many existing papers may similarly use such a strategy, but in a different form and less complicated way (of course due to their improper design, they cannot remove the data heterogeneity in the end).
>
> Following this statement, the reviewer would like to require stronger empirical experiments:
> * compare with other SOTA methods;
> * fair comparison e.g., also report results under the same amount of communication (indeed due to a different evaluation protocol, it is very hard for the reviewer to compare these results with the prior work);
> * larger experiments scale for deep learning tasks (current 25 nodes on a synthetic dataset is a bit trivial).

---

> > ### Author Response · Authors · 2022-11-23
> > **Reply to  i18n**
> >
> > ## Comparison with RelaySum
> > > compare with other SOTA methods
> >
> > We conducted the experiments of RelaySum and RelaySum with momentum (RelaySumM) for CIFAR-10 with VGG.
> > We show the results in the following table.
> >
> >
> >
> > | CIFAR-10 + VGG    | $10$-class | $2$-class |
> > | -------- | -------- | -------- |
> > | DSGD              | $89.7 \pm 0.15$      | $70.3 \pm 2.73$ |
> > | Gradient Tracking | $88.1 \pm 0.19$      | $82.9 \pm 0.13$ |
> > | RelaySum          | $88.3 \pm 0.13$      |  $86.3 \pm 0.15$ |
> > | DSGDm             | $\bf{92.2 \pm 0.09}$ | $39.6 \pm 5.92$ |
> > | QG-DSGDm          | $92.0 \pm 0.04$      | $77.8 \pm 1.96$ |
> > | DecentLaM         | $92.1 \pm 0.09$      | $85.2 \pm 0.67$ |
> > | RelaySumM         | $91.6 \pm 0.03$      |  $\bf{89.3 \pm 0.76}$ |
> > | Momentum Tracking | $91.9 \pm 0.06$      |  $87.0 \pm 0.48$ |
> >
> > The results indicate that RelaySumM is experimentally more robust to the data heterogeneity than Momentum Tracking
> > and can outperform Momentum Tracking when the data distributions are heterogeneous.
> >
> > However, Momentum Tracking has a clear advantage over RelaySum/RelaySumM in that the convergence rate of Momentum Tracking is proven to be independent of the data heterogeneity even when the momentum is applied.
> > The main objective and contribution of our study are not to achieve the SOTA, but to propose a method with momentum whose convergence rate is proven to be independent of the data heterogeneity in the standard deep learning setting.
> > We believe that our proof is valuable and helpful for future research that will attempt to analyze the convergence rates when the momentum is applied (e.g., the convergence rate of RelaySumM) regardless of whether the SOTA is achieved or not.
> >
> > We will add this experimental result and discussion in the revised manuscript.
> >
> > ## Fair Comparison
> >
> > > fair comparison e.g., also report results under the same amount of communication
> >
> > We respectfully disagree with this reviewer's statement.
> > In our experiments, all comparison methods were fairly evaluated by setting the total number of epochs to the same for each comparison method.
> >
> > We assume the reviewer is pointing out that the communication cost for Momentum Tracking is twice as much as for DSGDm, QG-DSGDm, and DecentLaM.
> > However, the objective of our experiments is to confirm that Momentum Tracking is also experimentally more robust to the data heterogeneity than DSGDm, QG-DSGDm, and DecentLaM, whose convergence rates are proven to depend on the data heterogeneity.
> > Comparison from the perspective of the communication costs is beyond our current scope and will be considered in future work.
> >
> > > indeed due to a different evaluation protocol, it is very hard for the reviewer to compare these results with the prior work
> >
> > We assume that the reviewer points out that Dirichlet distribution is more commonly used than the $k$-class setting used in the experiments to change the data heterogeneity.
> > However, as we replied in the previous response,
> > the experimental results indicate that Momentum Tracking is more robust to the data heterogeneity than DSGDm, QG-DSGDm, and DecentLaM in both cases where we use Dirichlet distributions and where we use $k$-class setting.
> > Additionally, we conducted the experiments in the case where we use Dirichlet distribution with $\alpha = 10.0$ (i.e., homogeneous setting).
> >
> >
> > | CIFAR-10 + VGG    | $\alpha = 10.0$ | $\alpha = 0.1$ |
> > | -------- | -------- | -------- |
> > | QG-DSGDm          | $89.8 \pm 0.08$      | $87.2 \pm 1.54$ |
> > | DecentLaM         | $90.3 \pm 0.28$      | $88.6 \pm 0.99$ |
> > | Momentum Tracking | $\bf{90.5 \pm 0.06}$ | $\bf{90.2 \pm 0.37}$ |
> >
> > We will add these experimental results in the revised manuscript.

---

> ### Author Response · Authors · 2022-11-30
> **Thank you for constructive feedback and for raising the score**
>
> We thank Reviewer i18n for constructive feedback and for raising the score from 5 to 6.
> As we replied, we promise to add the experimental results shown in the previous response in the revised manuscript.

---

### Comment · Area_Chair_Dpe5 · 2022-11-22
**Please respond as soon as possible if you still have questions on the paper.**

Please respond as soon as possible if you still have questions on the paper.

---

> ### Comment · Area_Chair_Dpe5 · 2022-11-29
> **Please respond to the authors by Nov. 30**
>
> Hi Reviewers i18nT and hspD,
>
> Please indicate whether the authors' rebuttal addresses your concerns.
>
> If you still have questions, please ask as soon as possible.

---

> > ### Comment · Area_Chair_Dpe5 · 2022-12-05
> > **Zoom Meeting**
> >
> > For all reviewers, which have not responded to the authors, I will have to ask you to meet via Zoom. If you want to avoid such an additional step, please respond by Dec. 5.

---

### Decision · Program_Chairs · 2023-01-20

**Decision:**

Reject

**Justification For Why Not Higher Score:**

NA

**Justification For Why Not Lower Score:**

NA

**Metareview: Summary, Strengths And Weaknesses:**

This paper introduces a momentum tracking approach for decentralized learning. The authors prove a theorem demonstrating that the proposed method is insensitive to data heterogeneity. They compare momentum tracking to other baseline methods, including gradient tracking, DSGDm, QG-DSGDm, and DecentLaM, through experimental evaluation.

The reviewers raised concerns on technical novelty and the significance of the results, especially on the dependence factor of hetergeneity. The concerns remain after the discussion. The paper can be significantly improved if it highlights the technical contributions and make in-depth comparison with existing work.